# Mapping Atlantic rainforest degradation and regeneration history with indicator species using convolutional network

Fabien H. Wagner[1,2]*, Alber Sanchez[1], Marcos P. M. Aidar[3], André L. C. Rochelle[4], Yuliya Tarabalka[5,6], Marisa G. Fonseca[1], Oliver L. Phillips[7], Emanuel Gloor[7], Luiz E. O. C. Aragão[1,8]

**1** Remote Sensing Division, National Institute for Space Research - INPE, São José dos Campos, SP, Brazil, **2** Geoprocessing Division, Foundation for Science, Technology and Space Applications - FUNCATE, São José dos Campos, SP, Brazil, **3** Department of Plant Physiology and Biochemistry, Institute of Botany, São Paulo, Brazil, **4** Center for Earth System Science, National Institute for Space Research - INPE, São José dos Campos, SP, Brazil, **5** Inria Sophia Antipolis, Sophia Antipolis, France, **6** Luxcarta Technology, Parc d'Activité l'Argile, Mouans Sartoux, France, **7** Ecology and Global Change, School of Geography, University of Leeds, Leeds, England, United Kingdom, **8** College of Life and Environmental Sciences, University of Exeter, Exeter, England, United Kingdom

* wagner.h.fabien@gmail.com

**Data Availability Statement:** The Worldview images and the 1962 aerial images are owned by DigitalGlobe and Emplasa, respectively, and cannot

## Abstract

The Atlantic rainforest of Brazil is one of the global terrestrial hotspots of biodiversity. Despite having undergone large scale deforestation, forest cover has shown signs of increases in the last decades. Here, to understand the degradation and regeneration history of Atlantic rainforest remnants near São Paulo, we combine a unique dataset of very high resolution images from Worldview-2 and Worldview-3 (0.5 and 0.3m spatial resolution, respectively), georeferenced aerial photographs from 1962 and use a deep learning method called U-net to map (i) the forest cover and changes and (ii) two pioneer tree species, *Cecropia hololeuca* and *Tibouchina pulchra*. For *Tibouchina pulchra*, all the individuals were mapped in February, when the trees undergo mass-flowering with purple and pink blossoms. Additionally, elevation data at 30m spatial resolution from NASA Shuttle Radar Topography Mission (SRTM) and annual mean climate variables (Terraclimate datasets at ~ 4km of spatial resolution) were used to analyse the forest and species distributions. We found that natural forests are currently more frequently found on south-facing slopes, likely because of geomorphology and past land use, and that *Tibouchina* is restricted to the wetter part of the region (southern part), which annually receives at least 1600 mm of precipitation. *Tibouchina pulchra* was found to clearly indicate forest regeneration as almost all individuals were found within or adjacent to forests regrown after 1962. By contrast, *Cecropia hololeuca* was found to indicate older disturbed forests, with all individuals almost exclusively found in forest fragments already present in 1962. At the regional scale, using the dominance maps of both species, we show that at least 4.3% of the current region's natural forests have regrown after 1962 (*Tibouchina* dominated, ~ 4757 ha) and that ~ 9% of the old natural forests have experienced significant disturbance (*Cecropia* dominated).

be shared publicly. However, a code example of the U-net model with a simulated Worldview image for reproducibility is available (https://doi.org/10.5281/zenodo.3601503) and the data of the maps presented in the Fig 8 are available on a public repository (https://doi.org/10.5281/zenodo.3601487). The authors confirm they had no special access or privileges to the Worldview images that other researchers would not have.

**Funding:** The research leading to these results received funding from the project BIO-RED 'Biomes of Brazil – Resilience, Recovery, and Diversity', which is supported by the São Paulo Research Foundation (FAPESP, 2015/50484-0) and the U.K. Natural Environment Research Council (NERC, NE/N012542/1). F.H.W. has been funded by FAPESP (grant 2016/17652-9). A.S. acknowledges the support of the FAPESP (grant 2016/03397-7). M.P.M.A. has been funded by ECOFOR Project, BIOTA-FAPESP (grant number 2012/51872-5). A.L.C.R. has been funded by supported by the State of São Paulo Research Foundation/FAPESP as part of the project ECOFOR (Process number 2012/51872-5) within the BIOTA/FAPESP Program - The Biodiversity Virtual Institute (www.biota.org.br) and co-supported by the British Natural Environment Research Council/NERC (NE/K016431/1). M.G.F acknowledges the support of Capes through a postdoctoral fellowship. Y.T. has been funded by the project EPITOME ANR-17-CE23-0009 of the French National Research Agency (ANR). L.E.O.C.A. thank the support of FAPESP (grant 2013/50533-5) and CNPq (grant 305054/2016-3). We also thank the Amazon Fund through the financial collaboration of the Brazilian Development Bank (BNDES) and the Foundation for Science, Technology and Space Applications (FUNCATE) no. 17.2.0536.1 (Environmental Monitoring of Brazilian Biomes). The funders (FAPESP, NERC, Capes, ANR and Luxcarta Technology) provided support in the form of salaries for authors F.W., A.S., Y.T. and M.G.F., but did not have any additional role in the study design, data collection and analysis, decision to publish, or preparation of the manuscript. The specific roles of these authors are articulated in the 'author contributions' section.

**Competing interests:** The authors have read the journal's policy and have the following conflicts: YT is employed by Luxcarta Technology. This does not alter our adherence to all the PLOS ONE policies on sharing data and materials.

## Introduction

Brazil holds 20% of Earth's biodiversity, and its third largest biome, the Atlantic rainforest, is a biodiversity hotspot and a global priority for conservation [1–4]. Over the two last centuries, this biome has been drastically reduced to less than 15% of its original area [5, 6]. Currently, the Atlantic forest is extremely fragmented with over 80% of the fragments < 50 ha and is poorly protected as reserves protect only 9% of the remaining forest, which represents ∼ 1–2% of the original area covered [7]. On the other hand, large areas of this biome are currently recovering from deforestation, as seen by an increase in tree cover since the year 2000 [8]. This increase in tree cover is driven mainly by eucalyptus plantations and natural regeneration [9], and also by forest restoration initiatives [10]. The natural regeneration, which occurs mainly in abandoned pasture lands, increases the provision of ecosystem services and habitat availability [11]. However, regional scale indicators to assess recovery stage, diversity or disturbance levels of the current natural forests are still underdeveloped, thereby adding uncertainty to the estimation of the ecosystem services and their value for conservation [12–14]. In this context, remote sensing is a key tool to monitor biodiversity, resources, and ecosystem services, as well as the human impact on natural ecosystems at a regional/biome scale [15–18].

Recently, a deep learning method called U-net was used with very high resolution images (WorldView-3, 30 cm spatial resolution, Digital Globe) to map natural and planted forests and to produce the first regional map of all individual adults of a natural tree species, *Cecropia hololeuca*, in a region of 1600 km$^2$ of the Atlantic forest in the state of São Paulo [14]. The classification accuracies above 95% for vegetation type and over 97% for *C. hololeuca* show that U-net outperforms other image segmentation methods and could support species mapping at regional scale [14]. Since *C. hololeuca* is a pioneer species, the spatial distribution of the individuals was then used to estimate anthropogenic disturbances inside the natural forest fragments, which are not accessible by others means. Over this same region, we have now gathered several other available high or very-high resolution remote sensed datasets for use in association with deep learning results or models to support conservation efforts of the Atlantic forest, to evaluate humans impacts and to understand forest landscape history.

In 1962, ten years before the launch of the first Landsat satellite, panchromatic aerial photographs were taken over São Paulo State by the company Aerofoto Natividade Ltda, including the region studied by [14], for the Brazilian Institute of Agronomy (Instituto Agronômico, IAC) at a spatial scale of approximately 1/25000. This benchmark of the forest cover will be used in this study to estimate the change in natural forest cover between 1962 and today (obtained from U-net), to assess the relative age of the forest (before or after 1962), and also to understand the currently growing planted forests cover, as the plantation of eucalyptus at large scale in this region only began after 1966, when the Brazilian federal law 5.106 came into force with the goal to encourage afforestation and reforestation in Brazil [19].

Additionally, very high resolution WorldView images will be used for the same region studied by [14], to map another pioneer tree, the species *Tibouchina pulchra*. This species is locally named *Manacá da Serra* and is mainly distributed in the Serra do Mar, a mountain system that follows the Atlantic coast where the largest remaining fragments of Atlantic forest are located. This tree species has ideal characteristics to support remote sensing methods of disturbance estimation in the Atlantic forest; it is found mainly in disturbed and secondary forests, sometimes being the dominant species [20–23]. Its trees bloom synchronously in February, and these crowns covered with pink flowers can be visually detected in very high resolution satellite images. The combined map of *Tibouchina pulchra* and *Cecropia hololeuca* is useful to understand the forested landscape of this Atlantic forest region.

Furthermore, high resolution climate and elevation data are available for this region. These datasets, along with tree species maps could bring new insights to species distribution models based on remote sensing data [17]. For example, TerraClimate is a new dataset with a high spatial resolution (1/24˚, ∼4km) of monthly climate and climatic water balance for global terrestrial surfaces from 1958-2015 [24]. Due to its high resolution, it reproduces in detail the strong climate gradient over the studied region. From the Atlantic coast following North through the Serra do Mar mountains and descending to the valley ("vale do Paraiba"), in less than 80 km, precipitation decreases from ∼ 2500–3000 mm.year$^{-1}$ to less than 1400 mm.year$^{-1}$ [24]. As *Tibouchina pulchra* is known to be sensitive to precipitation, its accurate location associated with climate data could help to assess its spatial distribution envelope. The region's topography also varies strongly in elevation (from 0 to 1650 m), slope, orientation and relief. With the elevation data at 30 m spatial resolution from the Shuttle Radar Topography Mission [25], indices could be derived to help describe and interpret the distribution of *Tibouchina pulchra* and *Cecropia hololeuca*. As the U-net method maps individual trees at an unprecedented scale, combining this information with climatic and edaphic conditions should bring new information on the species' preferences and distribution.

The following are the main objectives of this study. First, to assess the forest history in a sub-region of Atlantic forest ("Rio do Chapeu" region) based on the natural forest map established from a 1962 aerial image and the very high resolution natural/planted forest map of 2017 produced with the U-net [26]. Second, to produce the map of *Tibouchina pulchra* and to combine this map with the previously produced map of *Cecropia hololeuca* to analyse the relations of these two pioneer species with the history of the forest. Then, we use the obtained results to assess the anthropogenic disturbance history of the entire region. Finally, we analyse the association of *Tibouchina pulchra* and *Cecropia hololeuca* spatial distributions with climatic and edaphic conditions.

A more general objective is to show that with the new information on tree species locations brought by deep learning, new knowledge can be gained on the history of the forests and tree species distribution, even for the most visible and common tree species.

## Materials

### Study site

This study was undertaken in a region of the Atlantic forest biome located in the São Paulo State, Brazil, and centered at 23˚11'43'S and 45˚21'50W, Fig 1. The area was chosen because it contains several remnants of the Atlantic Forest biome as well as secondary forests at different stages of regeneration and planted eucalyptus forests, Fig 1b. Most of the forest plots from the BIOTA project [2], established to study the effect of fragmentation, are also located inside this region.

### WorldView-2 images and pre-processing

The two WorldView-2 images (DigitalGlobe, Inc., USA) were acquired over the region on February 17, 2017, at an average off nadir view angle of 18 and 20.7˚, respectively. DigitalGlobe catalog IDs of the images were A01001032124B900 and A01001032124BC00. These two images were distributed in tiles of 16384 × 16384 pixels which represents 30 tiles for each image. Only 36 tiles covering land were kept in the analysis covering a region of ∼ 2410 km$^2$, Fig 1. The spatial resolution was 0.5 m for the panchromatic band (464–801 nm) and 2 m for the selected multispectral bands: Red (629–689 nm), Green (511–581 nm) and Blue (447–508 nm). The three WorldView-3 images (DigitalGlobe, Inc., USA) over the region were acquired on August 13, 2017, at an average off nadir view angle of 20.1˚, further details are given in [14].

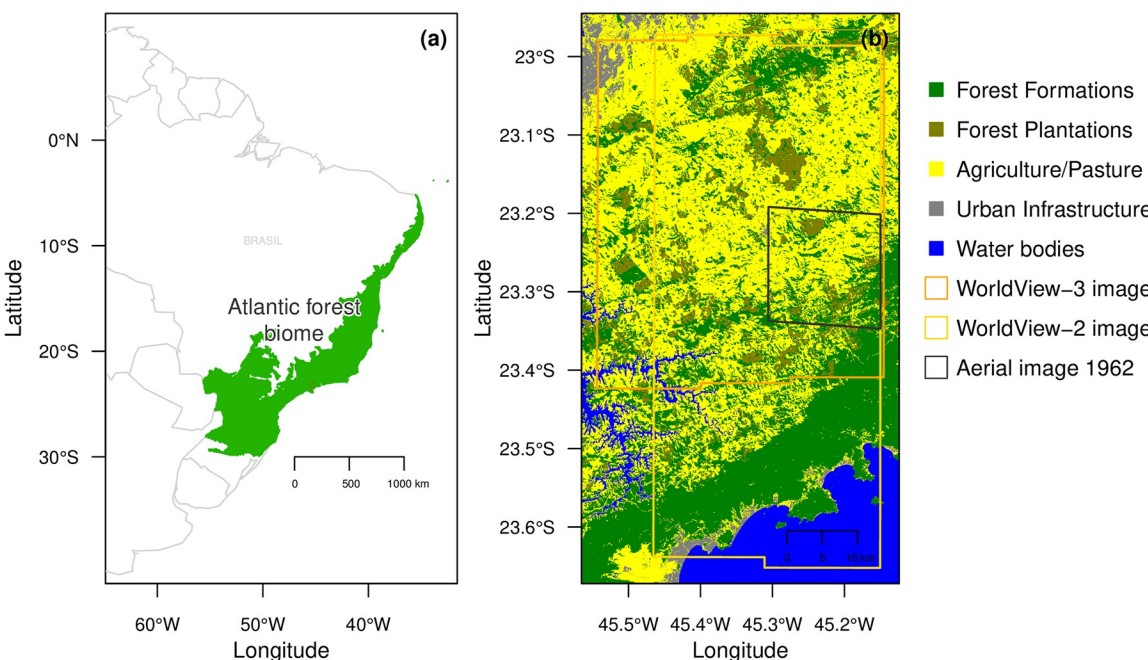

**Fig 1.** Geographical locations of the Atlantic Forest biome in green and of our region of interest in red (a); region of interest with 2017 land use/cover classes from the MapBiomas project, the WorldView-2 and WorldView-3 tiles used in this study, and the area covered by the aerial photography of 1962 called further "Rio do chapeu" region (b).

All bands were scaled from raw image digital numbers (11 bits) to 0–255 (8 bits). The Red-Green-Blue (RGB) bands were pan-sharpened with the panchromatic band using the method Simple RCS of the Orfeo toolbox addon `otbcli_BundleToPerfectSensor` [27] to create a single high-resolution RGB image at 0.5 m spatial resolution. No atmospheric correction was performed.

## Forest cover in 1962

To assess the recent history of the current forest stands and to test how the pioneer tree species *Tibouchina pulchra* and *Cecropia hololeuca* were associated with the regeneration of abandoned pasturelands or disturbed forest, we used an independent dataset of aerial photographs taken in 1962 over the São Paulo State region. The aerial photographs were realised in 1962 by the company Aerofoto Natividade Ltda for the Brazilian Institue of Agronomy (Instituto Agronômico, IAC) at a scale of approximatelly 1/25000 and made available by the Brazilian Institute of Geography and Cartography (Instituto Geográfico e Cartográfico do Estado de São Paulo, IGC). The IGC data are not freely available but low resolution samples can be browsed at http://datageo.ambiente.sp.gov.br/app/# in the directory `/Base Imagem/Imagens IGC/ IGC—Fotoíndice Vôo 1962`. The sample contained 40 aerial photographs covering the region of the "Rio do Chapeu" near São Luiz do Paraitinga (Fig 1b), with an overlap of approximately 40% to 50% between each adjacent photography. The images were mounted in two panoramas using the software AutoStitch© [28, 29]. Then, these two panoramas were registered using the 2017 WorldView-2 images in ArcGis 10.4 [30] with 290 and 343 control points, respectively, using a spline function for interpolation.

## Climate data

To test if the *T. pulchra* and *C. hololeuca* spatial distributions were related to local climate, we computed the mean annual values of 12 variables from the high-resolution global dataset Terraclimate at $\sim$ 4 km spatial resolution (2.5 arc-minute, 1/24th degree) [24]. The climate variables used were actual evapotranspiration (mm), climatic water deficit (mm), Palmer drought severity index, reference evapotranspiration (mm), precipitation (mm), soil moisture (mm), downward shortwave radiation ($Wm^2$), maximum temperature (˚C), minimum temperature (˚C), vapor pressure (KPa), vapor pressure deficit (kPa) and windspeed ($m.s^{-1}$) [24].

## Elevation data

To test if the *T. pulchra* and *C. hololeuca* spatial distributions were related to elevation (or elevation-related variables), elevation data from the shuttle radar topography mission at 30 m spatial resolution were used [25]. From this dataset, we used the variable 'elevation' and computed five additional elevation related indices that consider the eight neighbor pixels: slope (˚); aspect, which indicates the compass direction that the slope faces (in ˚, where 0 is the North); topographic position index, which compares the elevation of the pixel to the mean elevation of the neighboring pixels (positive value indicates ridge, negative value indicates valley and near zero value indicates flat areas or areas of constant slope); terrain ruggedness index, an index of topographic heterogeneity computed as the sum of the change in elevation between the pixels and its eight neighbors cells (zero indicates no ruggedness); and, finally, roughness, computed as the difference between the maximum and the minimum value of a cell and its eight surrounding cells [31, 32].

## Natural and planted forests mask

Natural and planted forests are identifiable in the images by their color and structure, see Fig 2. All the forests were mapped in two tiles of the Worldview-2 image using the U-net algorithm and the weights previously trained to segment forests from Worldview-3 images [14]. After all forests were segmented in both tiles, planted forests were then manually delineated resulting in 2154 polygons (716 + 1438). Cropping the mask of the two tiles in 256 × 256 pixels images resulted in a sample of 7780 images to train the model of natural and planted forests detection and segmentation. Additionally, outside the extent of the Worldview-2 image, we used previous natural and planted forest maps produced with Worldview-3 images and the same U-net methodology used here [14]. For consistency with the map produced with the WorldView-2 image, these maps were resampled from 0.3 to 0.5 m spatial resolution with the function average and coded as 1 if the pixel value was above 0 (natural or planted forest in the 0.5 m pixel) or 0 otherwise.

## *Tibouchina pulchra* mask

All individuals of *T. pulchra* bloom synchronously in February, rendering the species identifiable in the images due to its unique rose/purple color, see Fig 3. All occurrences of *Tibouchina pulchra* trees were manually delineated in one tile of the Worldview-2 images. The delineated sample was composed of 10813 polygons where each polygon can represent more than one individual. With the delineated polygons, a raster mask coded in RGB was produced with these values: Background [0, 0, 0] and *Tibouchina pulchra* [127, 127, 127]. Cropping the mask in 128 × 128 pixel images resulted in 1002 images containing at least one *T. pulchra* tree for the training sample. Additionally, 1046 randomly sampled images of the same tile containing

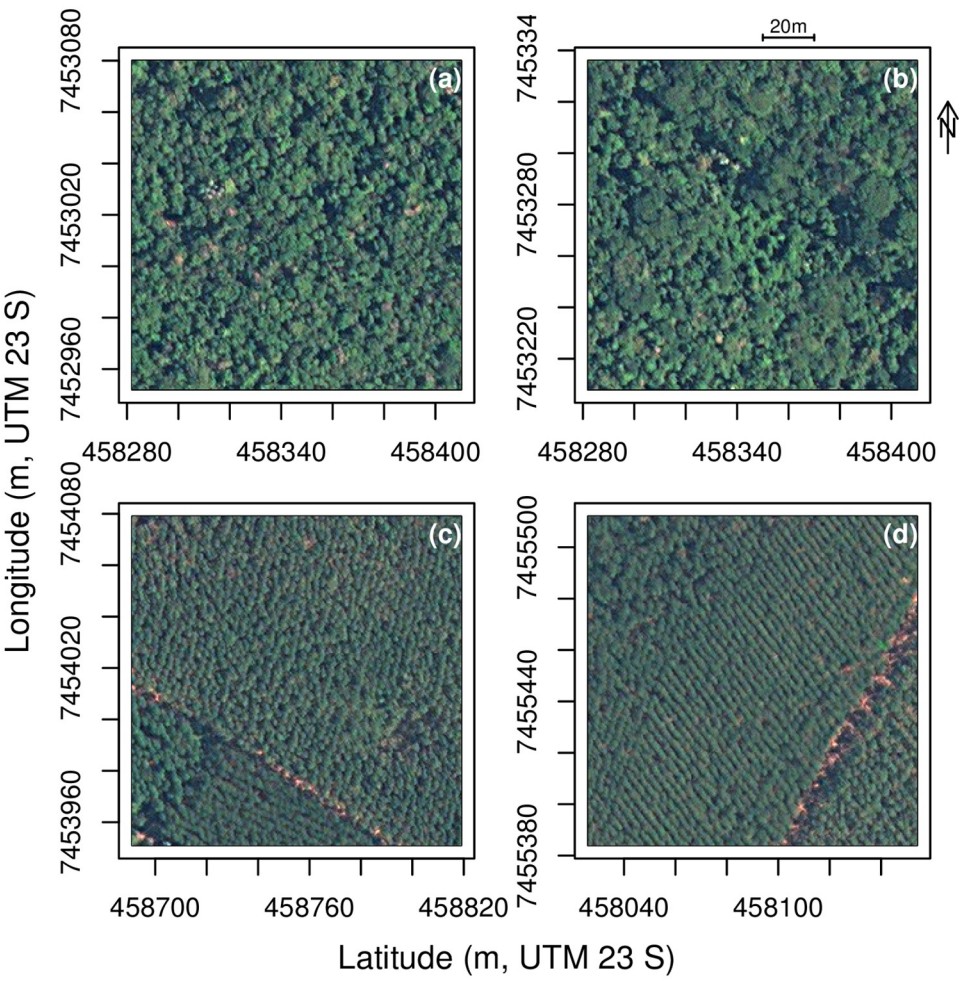

**Fig 2. Example of images (256 × 256 pixels) with natural (a-b) forest and planted eucalyptus forest (c-d).** Satellite image courtesy of the DigitalGlobe Foundation.

only background were added to the training set to provide a wider range of images to the network with features such as constructions or pastureland.

## Cecropia map

For the *Cecropia* trees in the region, we used a map previously produced in [14] with Worldview-3 images and the same U-net methodology used here for *Tibouchina pulchra*. For consistency with the map produced with the WorldView-2 image, the *C. hololeuca* map was resampled from 0.3 to 0.5 m spatial resolution with the function average and coded as 1 if the pixel value was above 0 (presence of *Cecropia* in the 0.5 m pixel) or 0 otherwise.

## Methods

### U-net model

**Architecture.** In this study, we used a convolutional network for multi-class image segmentation known as U-net [14, 26]. This network performs a per-pixel classification, predicting the probability of each pixel to belong to a particular class. This U-net model has recently

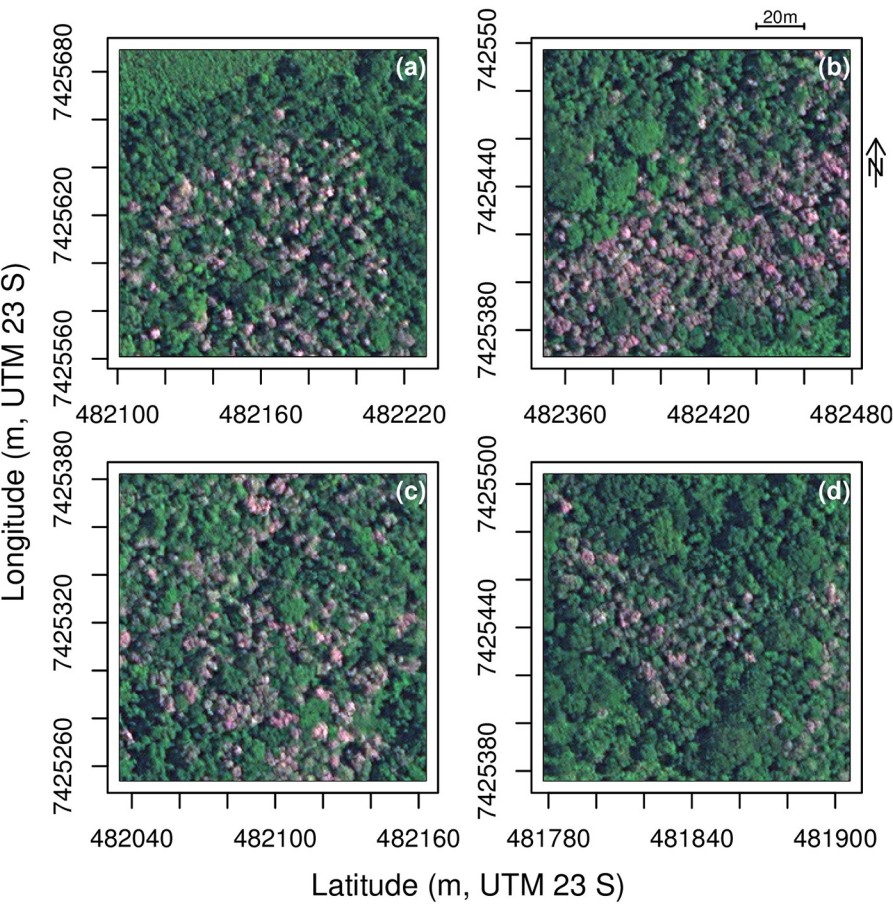

**Fig 3. Example of images (256 × 256 pixels) with the species of interest, *Tibouchina pulchra* (a-d).** Satellite image courtesy of the DigitalGlobe Foundation.

proven to become a new standard in image dense labeling [33]. We adapted the U-net architecture [26] with twice less filters, since our training set is limited and a smaller number of filters helps in preventing overfitting. Furthermore, we used a three-band RGB image as the input and have adapted the network architecture accordingly, see [14] and Fig 4. Sigmoid activation functions were used to ensure that output pixel values range between 0 and 1. For the training, we used an input size of 256 × 256 pixels for forest type segmentation and 128 × 128 pixels for the *Tibouchina pulchra* trees segmentation.

**Network training.** The training samples comprised 2048 images of 128 × 128 pixels for *T. pulchra* trees and 7780 images of 256 × 256 pixels for forest types. The size of 128 × 128 pixels was selected because *T. pulchra* crowns are smaller than 128 pixels in diameter (128 pixels = 64m). An image size of 256 × 256 pixels was used for natural forest/plantations to include better contextual; plantation trees are generally planted in lines, and this information is more noticeable with this image size. The images were extracted from uniform grids of 128 × 128 pixels and 256 × 256 pixels, without any overlap between neighboring images. 80% of these images were used for training, and 20% used for validation. During network training, we used a standard stochastic gradient descent optimisation. The loss function was designed as a sum of two terms: mutual cross-entropy and Dice coefficient-related loss [34–36]. We used the optimiser RMSprop (unpublished, adaptive learning rate method proposed by Geoff Hinton here http://www.cs.toronto.edu/~tijmen/csc321/slides/lecture_slides_lec6.pdf) with an

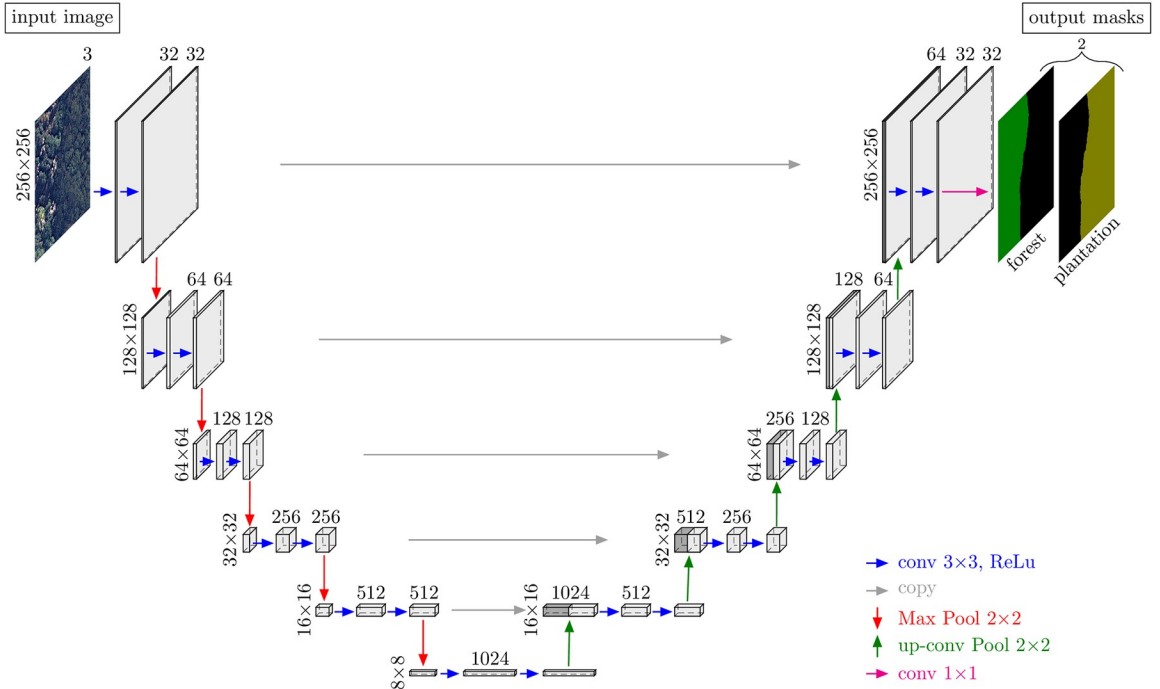

**Fig 4. U-net architecture for the forest types segmentation, adapted from [26] and reproduced from [14].** The number of channels is indicated above the cuboids and the vertical numbers indicate the row and column size in pixels.

initial learning rate of 1e-4. We trained our network for 100 epochs, where each epoch comprised 78 batches with 16 images per batch. The optimisation was stopped when the loss function improvement did not exceed 1e-4.

Data augmentation was applied randomly to the input images, including 0/90/180/270° rotations and changes in the brightness, saturation and hue by converting RGB to Brightness-Saturation-Hue space (BSH) and modulating the current values between 95%–110% for brightness, 95%–105% for saturation and 99%–101% for hue (as changes in the plant hues are not expected).

**Segmentation accuracy assessment.** Three performance metrics were computed. First, the overall accuracy was computed as the percentage of correctly classified pixels. Second, the intersection over union (*IoU*) of the object class, which is the number of pixels labelled as object in both the prediction and the reference, divided by the number of pixels labelled as object in the prediction and in the reference. Third, the F1 score was computed for each class *i* as the harmonic average of the precision and recall, Eq 1, where precision was the ratio of the number of segments classified correctly as *i* and the number of all segments (true and false positive) and recall was the ratio of the number of segments classified correctly as *i* and the total number of segments belonging to class *i* (true positive and false negative). This score varies between 0 (lowest value) and 1 (best value).

$$F1_i = 2 \times \frac{precision_i \times recall_i}{(precision_i + recall_i)} \tag{1}$$

**Prediction.** For prediction, each Worldview-2 tile of 16384 × 16384 pixels was cropped based a regular grid of 512 × 512 pixels, and 64 neighbour pixels were added on each side to

create an overlap between the patches. If there was a remaining blank portion (for example, due to the tile border) it was filled by the symmetrical image of the non blank portion. The prediction of both models (forest types and *T. pulchra* tree segmentations) were made on these images of $640 \times 640$ pixels, and the resulting images were cropped to $512 \times 512$ pixels and merged to reconstitute the $16384 \times 16384$ pixels WV-2 tile. This overlapping method was used to avoid the artifact of prediction on the border, a known problem for the U-net algorithm [26]. To belong to a given class, the pixel prediction value must be greater than or equal to 0.5. From the raster image of the predictions, we generated the spatial polygons of each classes.

**Algorithm.** The model was coded in R language [37] with Rstudio interface to Keras [35, 36] and Tensorflow backend [38]. R code is available upon request. The training of the models took $\sim$ 2 to 20 hours using GPU on a Nvidia Quadro K6000 with a 12 GB dedicated memory. Prediction using GPU of a single tile of $16384 \times 16384$ pixels ($\sim 67$ km$^2$) took approximately 35 minutes. The code of the U-net model was adapted from the original U-net code developed for Keras and Rstudio, and is available here http://doi.org/10.5281/zenodo.3601503.

## Forest and species distribution analysis

**Past history of the 2017 forest cover.** First, the mask of natural/planted forest in 2017 was produced with the U-net at 0.5 m. Second, the past history of the 2017 forest cover, that is, if they were present or not in 1962, was mapped. To do this, only for the forests that were mapped in 2017, the mask of the forests in 1962 was produced with the registered aerial images of 1962. This map was produced in QGIS and resampled in a raster tile at 0.5 m spatial resolution [39]. The history of the 2017 forest cover was then described in the "Rio do Chapeu" region. With these two maps, we were able to verify if a forested pixel present in 2017 was already present in 1962 (old forest) or if it was a recent regrowth, posterior to 1962 (new forest). We also determined how much pasture and forest were converted to eucalyptus plantation. Finally, a raster mask at 0.5 m was created with three classes, eucalyptus plantation, old natural forest and new natural forest in 2017.

**Association of pioneer species with forest history.** The "Rio do Chapeu" region was covered by the 1962 images, and here we determined the association between the pioneer species and the age of the forest. We describe for each *Tibouchina pulchra* and *Cecropia hololeuca* pixels if they were inside an old or new natural forest (regrowth after 1962) and their minimum distance to new and old natural forest, within a maximum distance threshold of 25 m and of 50 m, respectively. To assess the dominance of both tree species in the natural forest, we computed the percent of pixel of the species in the forest in a circular area of 100 m radius around each pixel of the species; that is, the number of pixels of the species was divided by the number of pixels of the forest in the area. This index ranges from 0 to 100% and, for example, if it is equal to 50% this means that 50% of the canopy is occupied by the species. These indices were computed for both pioneer species in new and old forests. This index was computed for the "Rio do Chapeu region", then we used the mean of the index in new and old forest to determine, where both species were dominant, here defined as pixels with a higher compaction that the mean of the compaction of the species in the new or in old forest.

**Distribution of *Tibouchina pulchra* and *Cecropia hololeuca* in relation to elevation and local climate.** Both species' distribution associations with local climate (see section for more details) and with altitude and related topographic indices (see section) were described over the full extent of the Worldview-3 image for *C. hololeuca* and over the full extent of the WV2 image for *T. pulchra*, Fig 1. First, for a given species, the centroid coordinates of the delineated polygon were computed. Then, the same amount of points was sampled randomly over the map. The values for climate, elevation and related topographic indices were extracted for the

**Table 1. Numerical evaluation of the models, training and validation sample size and convergence details.**

| model | epoch | batch | training sample | validation sample | IoU | Overall accuracy |
|---|---|---|---|---|---|---|
| Forest types | 28 | 8 | 6224 | 1556 | 0.950 | 95.71 |
| Tibouchina trees | 44 | 16 | 1638 | 410 | 0.867 | 98.92 |
| Cecropia trees[a] | 28 | 16 | 1946 | 486 | 0.861 | 97.09 |

[a] from [14]

coordinates corresponding to the centroids (real species distribution) and to the random points (random distribution). Then the distributions of the variable at random coordinates and at the real species coordinates were compared using histograms. In the text, the values extracted from the random coordinates are referred to as mean landscape values.

## Results

### Detection accuracy

The time for convergence was ∼ 20hrs for the forest-types model and < 3hrs for the *T. pulchra* trees model. The best models were obtained after 28 epochs with 8 images per batch for the forest-type model and 44 epochs with 16 images per batch for the *T. pulchra* trees model, Table 1. For the forest type segmentation, the overall accuracies and *IoU* coefficient were 95.71% and 0.95, respectively. The overall accuracy of the *T. pulchra* trees segmentation was 98.92% and the *IoU* coefficient was 0.87. Accuracies of the *C. hololeuca* tree segmentation previously presented in [14] are similar to the accuracies of *T. pulchra* segmentation with an overall accuracy of 97.09% and an *IoU* coefficient of 0.861.

The natural forest class showed the best F1-score, followed by the eucalyptus plantation class, Table 2. Recall was higher in natural forests than in plantations, thereby indicating a lower rate of false negatives. Precision was higher in natural forests than in plantations, thereby indicating a lower rate of false positives. The F1-score of the *Tibouchina* trees segmentation was lower than forest-types F1-score, with a value of 0.87. For the *Tibouchina* class, precision was slightly higher than recall, with values of 0.88 and 0.85 respectively, Table 2. The F1-score of the *Cecropia* trees segmentation obtained previously was lower than forest-types F1-score, with a value of 0.80 [14]. For the *Cecropia* class, F1-score, precision and recall values were similar, Table 2.

Examples of *Tibouchina* segmentation result with F1-score and manual segmentation are presented in Fig 5. The crowns of *T. pulchra* trees are small, mainly < 10 m of diameter. The border of the *T. pulchra* crowns is not sharp in the image, and there is often a small variation

**Table 2. F1-scores of the segmentation for natural and eucalyptus forests as well as for *T. pulchra* and *C. hololeuca* trees.**

| model | classes | precision | recall | F1-score |
|---|---|---|---|---|
| Forest types | Natural forest | 0.956 | 0.956 | 0.956 |
|  | Eucalyptus plantation | 0.947 | 0.939 | 0.943 |
|  | background | 0.970 | 0.965 | 0.968 |
| Tibouchina trees | Tibouchina trees | 0.884 | 0.851 | 0.868 |
|  | background | 0.993 | 0.995 | 0.994 |
| Cecropia trees[a] | Cecropia trees | 0.808 | 0.801 | 0.804 |
|  | background | 0.985 | 0.983 | 0.984 |

[a] from [14]

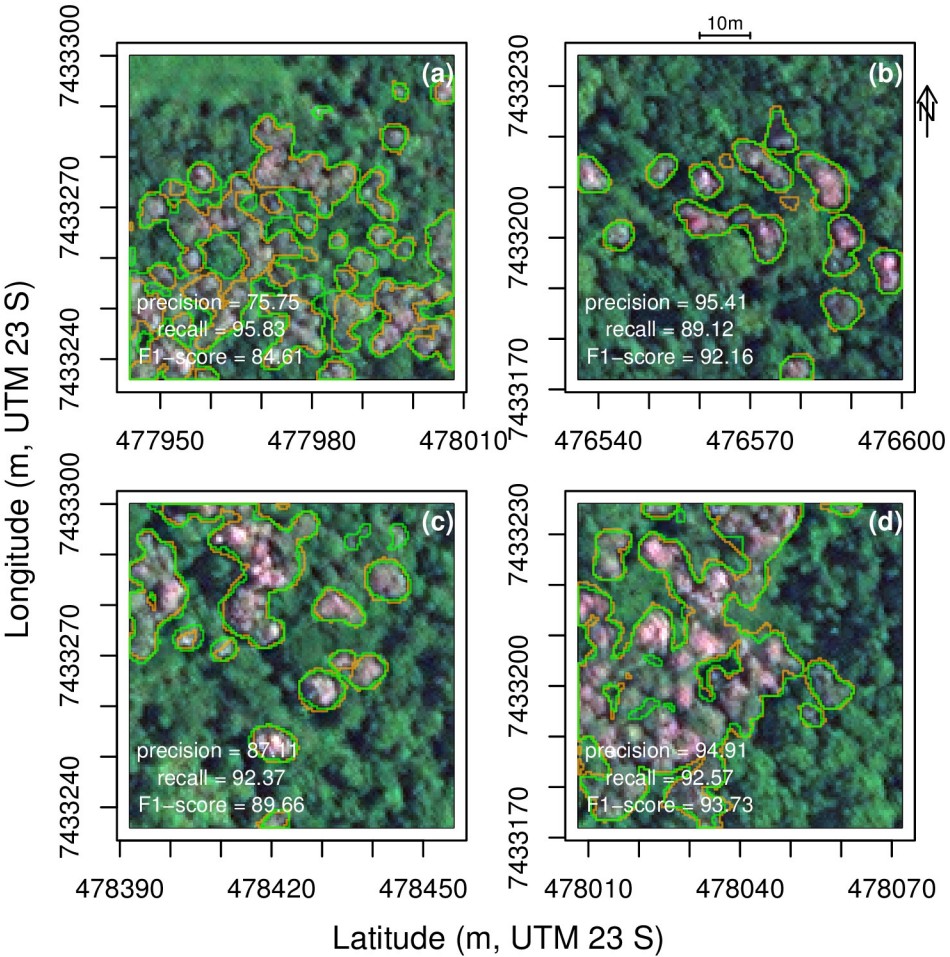

**Fig 5. Example of validation images (128 × 128 pixels) for *T. pulchra* trees with manual delineation in orange and the U-net delineation in green (a-d).** Satellite image courtesy of the DigitalGlobe Foundation.

between the manual and automatic segmentation, Fig 5. Some small *T. pulchra* trees were missed by the algorithm, Fig 5b, or missed by the producer, Fig 5c. When *T. pulchra* trees present larger crowns and/or concentrated patches, there is a very good agreement between manual and U-net segmentation Fig 5d.

Besides eucalyptus unnoticeable plantation structure and natural forests' homogeneous canopy [14], the most common source of error here was the detection of old planted forests. Old plantations that were well delineated with WV-3 data (0.3 m spatial resolution) presented more error and were more difficult to detect by the algorithm in WV-2 images (0.5 m spatial resolution). In these plantations, the plantation structure (regular lines of planted trees) has disappeared; the crowns were large and did not show regular size and shape.

## Forest cover and studied species distribution in the "Rio do Chapeu" region

**Forest cover in the "Rio do Chapeu" region.** In 2017, the forests covered 55.3% (142 km$^2$) of the "Rio do Chapeu" region, Fig 6. 43% of these forests were already present in 1962 (hereafter referred as old forests) while 57% of forests were pasture (hereafter referred as new

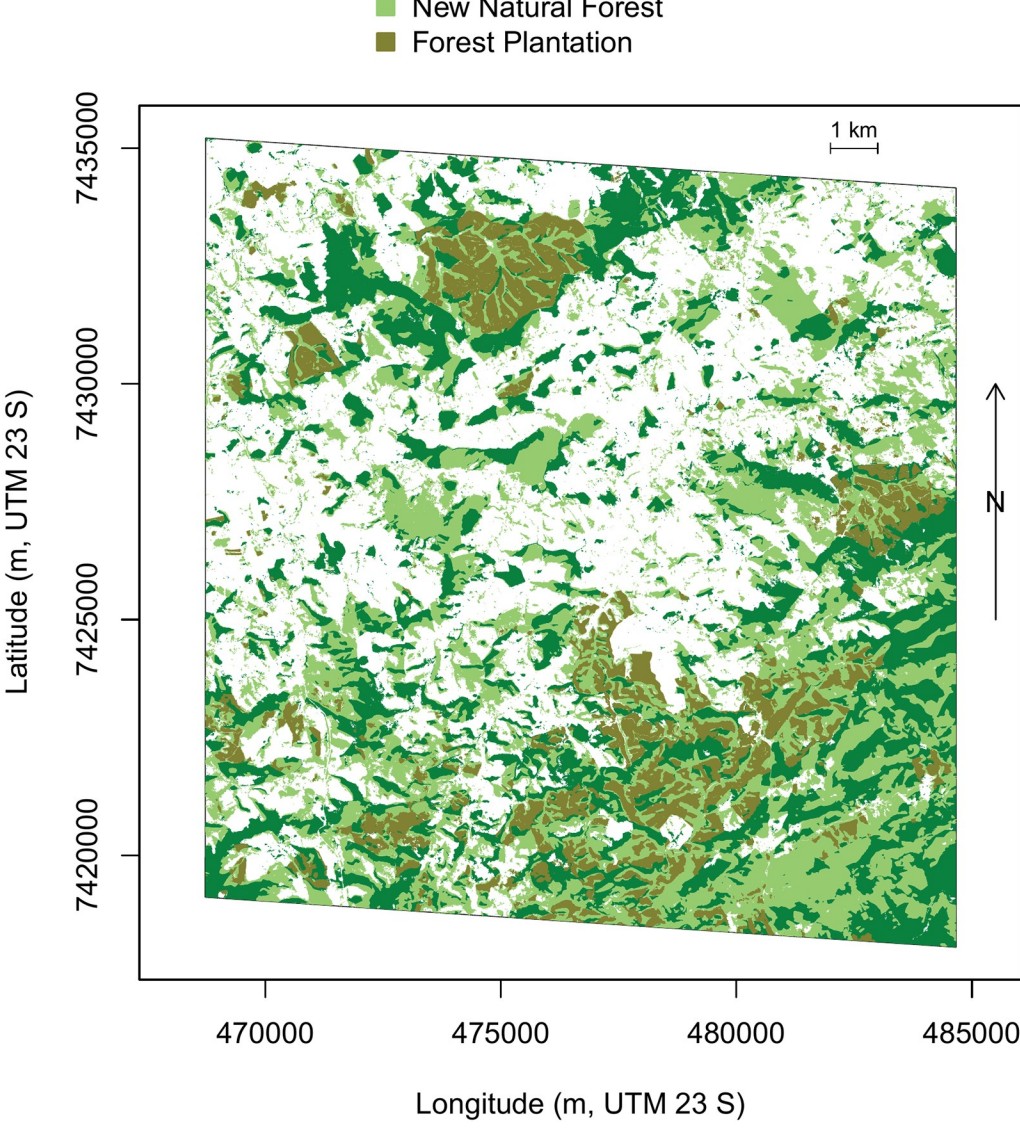

**Fig 6. Distribution and relative age of the "Rio do Chapeu" forests in 2017.** Two types of forests, planted or natural, were identified with the U-net method. And, among the natural forests, new forests indicate forests that were pasture in 1962 (from the registered 1962 aerial photos) while old forests indicate forests that were already there in 1962. Note that all eucalyptus forest have been planted after 1962. All the pixels which are not planted or natural forests are in white.

forests). Among the new forests, 76.9% were natural regeneration while 23.1% were eucalyptus plantation. The eucalyptus were mainly planted where there was no forest in 1962; only 21.1% of the plantation were natural forests in 1962. As only two time periods have been assessed, 1962 and 2017, intermediate land uses that could have existed between transition from natural forests and to planted forests are unknown.

**Cecropia hololeuca distribution pattern.** Cecropia covered 0.25% of the current natural forest, Fig 7. An interesting pattern is observed in their distribution: 75% of these trees are located in forests that were already established in 1962, 86% inside or at a distance less than 25 m from an old forest and 91% inside or at a maximal distance of 50 m from an old forest.

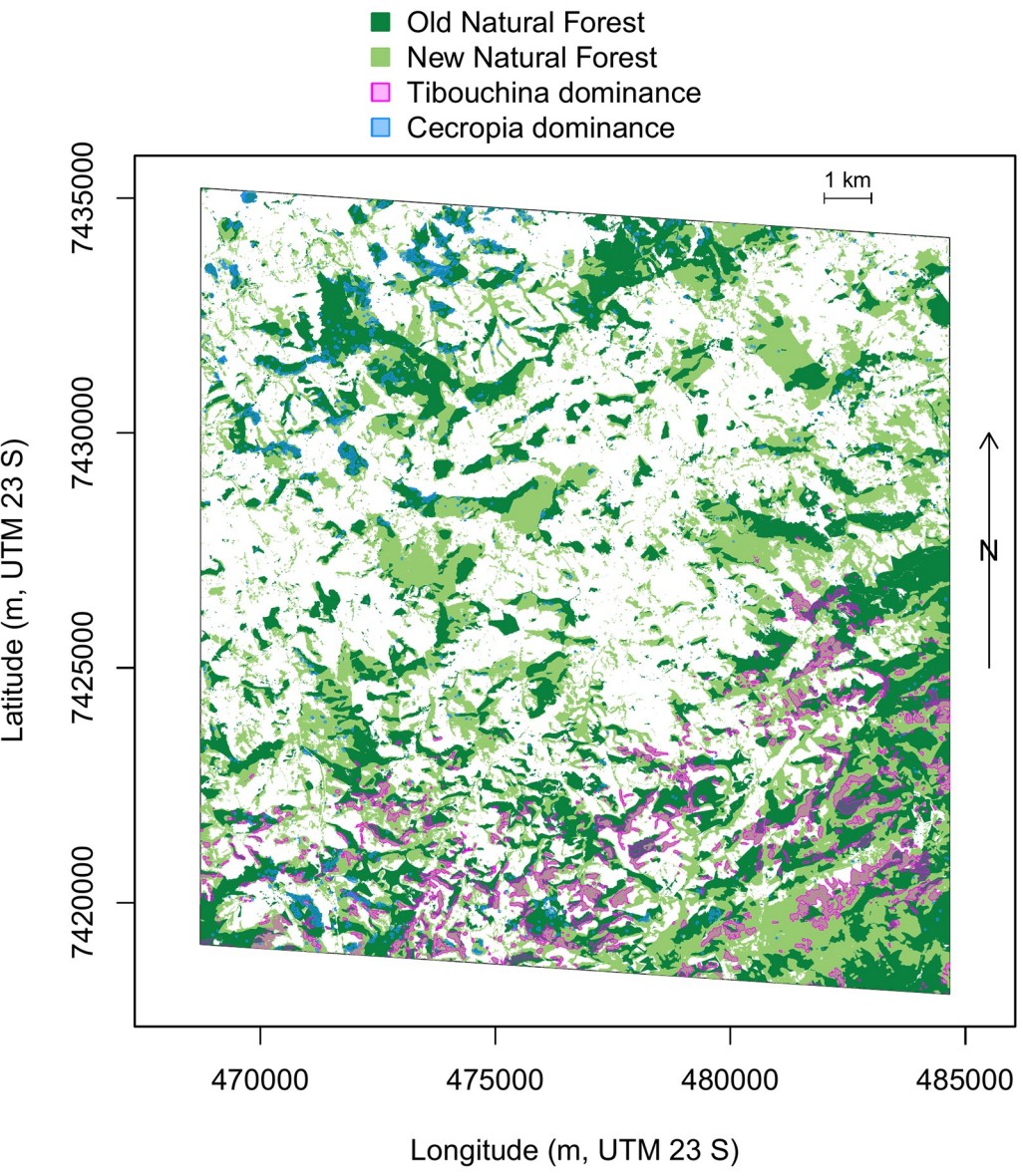

**Fig 7. Distribution of *Tibouchina pulchra* (transparent magenta) and *Cecropia hololeuca* (transparent blue) dominances in the natural forests of the "Rio do Chapeu" region.** Dominance was defined as pixels with more than 2.84% of *C. hololeuca* or more than 17.5% of *T. pulchra* within a radius of 25 m, for *C. hololeuca* and *T. pulchra*, respectively. New forests indicate forests that were pasture in 1962 (from the registered 1962 aerial photos) while old forests indicate forests that were already there in 1962. All the pixels which are not natural forests are in white.

Considering only the old forest, in the neighboring area of 25 m around each *C. hololeuca* pixel, there is a mean of 2.84% of *C. hololeuca* pixels (55.8 m$^2$); and 1.2% (94.25 m$^2$) in a neighboring area of 50 m. These low percentages indicate that *C. hololeuca* trees did not form homogeneous forest cover but were mixed with other trees species. We further use this threshold of 2.84% of *C. hololeuca* pixels to define area where there is *C. hololeuca* dominance (pixel with more than 2.84% of neighbors *C. hololeuca* pixels within a radius of 25 m).

**Tibouchina pulchra distribution pattern.** *Tibouchina pulchra* with flowers covered 3.4% of the current forest, Fig 7. For this species, an opposite behaviour of the *C. hololeuca* is observed. 66.8% of these trees were located in new forests that were pasture in 1962, 81.4%

inside or at a distance below 25 m of a new forest and 89.7% inside or at a distance below 50 m of a new forest. *T. pulchra* is also over-represented close to the "Serra do Mar", which is located in the bottom of "Rio do Chapeu" region. Considering only the new forests which contain the majority of the *Tibouchina* trees, in the neighboring area of 25 m around each *Tibouchina* pixels, there is a mean of 17.5% of *Tibouchina* pixels (343.5 m$^2$) and 12.9% (1011.3 m$^2$) in a neighboring area of 50 m. *Tibouchina*, therefore, is distributed in more homogeneous and dominant stands than the *C. hololeuca*.

## Species regional distibution

At the regional scale, forests with *T. pulchra* dominance are distributed almost exclusively in the South where the forest is fragmented and before the edge of the 'Serra dor Mar', Fig 8. Overall, forested pixels with *T. pulchra* dominance cover 4.28% of the natural forests in the WorldView-2 image ($\sim$ 4757 ha), however it can reach 21.89% in some tiles of the image. This species forms patches with a mean area of 0.27 ha and the maximum observed size is of 72.2 ha. Few small patches can be observed in the largest fragment of natural forest in the South, which is the largest remaining fragment of Atlantic forest. *C. hololeuca* is distributed all over the region, Fig 8, and forested pixels with *C. hololeuca* dominance covered 8.8% of the natural forests in the WorldView-3 image. In the 'Serra dor Mar', the area dominated by *C. hololeuca* and *T. pulchra* appeared systematically separated in space, Fig 8.

**Forest and species distribution associations with elevation and related variables.** Natural forests are present at all elevations, while non-forested area are limited to the sea level or to altitude above 500 m, Fig 9a. Areas with altitudes between 50 and 500 m are only present in the steep, forested slopes of the Serra do Mar (the first uphill). Forests are found more often on steeper slopes than non-forest areas, Fig 9b. Furthermore, the distribution of forest aspect indicates that they occur predominantly on south-oriented slopes (south is 180˚), Fig 9c. On the other hand, non-forested areas are more frequently oriented to the North (0˚). There is no apparent difference of TPI between the two classes, Fig 9d. Finally, forests are distributed on slightly rougher terrains than the non-forested area, Fig 9e and 9f. All these characteristics of the forest distribution remain even after removing the largest remaining fragment at the south, which could have biased the results, S1 Fig.

As forests have a particular association with elevation and related variables, to avoid misinterpretation, in the following we compare the distribution of the *T. pulchra* and *C. hololeuca* to the distribution of forested pixels without their dominance. The distributions of *T. pulchra* and *C. hololeuca* presented an important overlap with the distribution of natural forests without their dominance's, S2 and S3 Figs. They were both mainly found at elevations above 600 m with a peak of their distribution at 800-900 m S2a and S3a Figs. They are found similar slopes, even if the distribution of the slope for *T. pulchra* is more skewed, b and S3b Fig. As for the forest, they both occur predominantly on the south-oriented slopes, S2c and S3c Figs. For the topographic position there is a slight tendency of *T. pulchra* to appear on TPI more negative than the rest of the forest d, while the contrary is observed for the *C. hololeuca* S3d Fig. Finally, both species are distributed on terrains with similar roughness as other natural forested area, even if the distribution of roughness for *T. pulchra* is a little more skewed, S2e, S2f, S3e and S3f Figs.

**Species distribution associations with climate variables.** For most of the climate variable, *T. pulchra* distribution in relation to climate variables is different from a random distribution association with climate variable, that is, the mean climatic landscape, Fig 10. For the climate variables associated with water, precipitation and soil moisture, *T. pulchra* distribution was narrower than the distribution of the mean climatic landscape, peak at a median annual

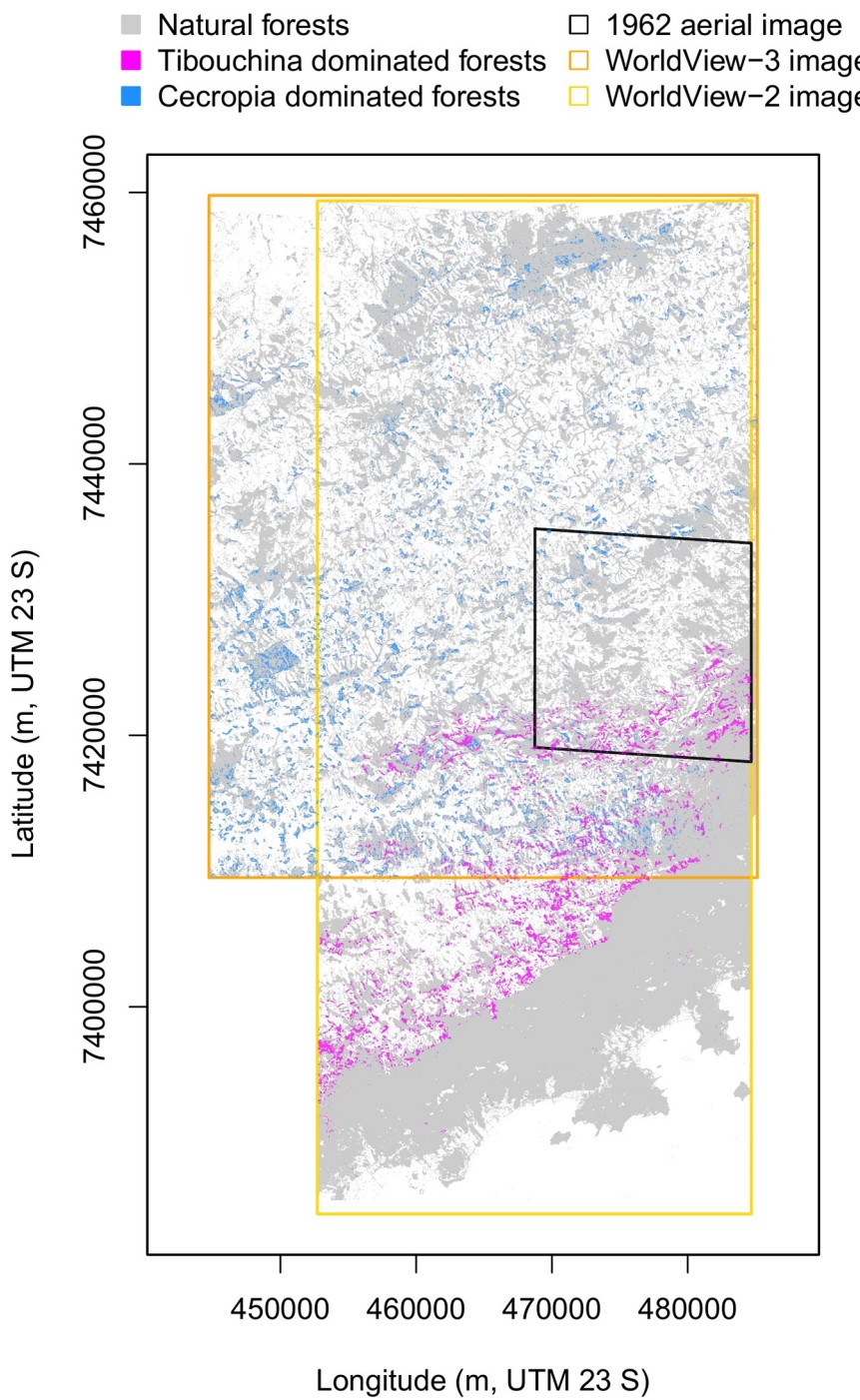

**Fig 8. Map of the natural forests in 2017 with additional information on *Tibouchina pulchra* and *Cecropia hololeuca* dominances.** Dominance was defined as pixels with more than 2.84% of *C. hololeuca* or more than 17.5% of *T. pulchra* within a radius of 25 m, for *C. hololeuca* and *T. pulchra*, respectively. Natural forest, *T. pulchra* and *C. hololeuca* were mapped on the extends cover by the Worldview-3 and -2 image.

precipitation of 1775 mm, Fig 10a, and occurred on soil with a higher soil moisture (peak at 670 mm versus 550mm), Fig 10b. Its reference evapotranspiration and actual evapotranspiration present a skewed distribution around 905 and 965, respectively, Fig 10c and 10d. Its distribution encompasses locations with lowest climatic water deficit and the distibution of Palmer

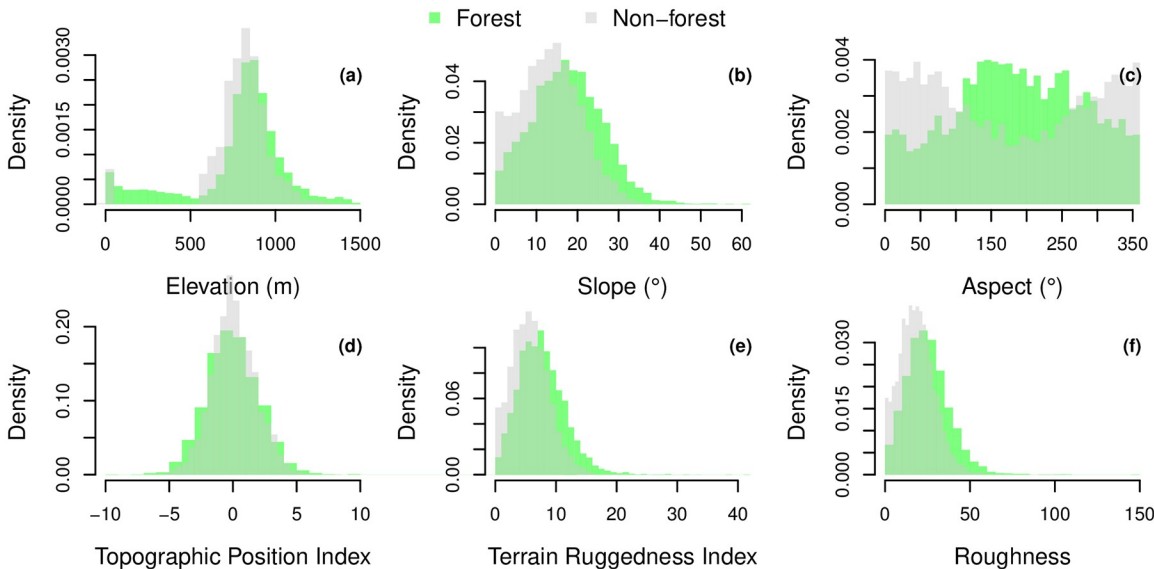

**Fig 9. Distribution of the natural forests and non forest (drawn from a random distribution) in relation to elevation and five elevation-related variables at the regional scale.** The five elevation-related indices consider the 8 neighbor pixels and are slope (˚); aspect (compass direction facing slope in ˚, where 0 is the North); topographic position index (elevation of the pixel in relation to the mean elevation of the neighbor pixels); terrain ruggedness index (sum of the change in elevation between the pixels and its 8 neighbors cells); and finally roughness (difference between the maximum and the minimum value of a cell and its 8 surrounding cells) [31, 32].

drought index peak at a value below the value of the mean climatic landscape, Fig 10e and 10f. For climate variables related to radiation, *T. pulchra* occurred in locations with less downward shortwave radiation, and maximum temperature lower that the mean climatic landscape, Fig 10g and 10h, while there is not so much difference for minimum temperature, Fig 10i. As for minimum temperature, not much differences are observed for the vapor pressure, Fig 10j, while *T. pulchra* occurred in location with lowest vapor pressure deficit than mean climatic landscape, Fig 10k. Finally, *T. pulchra* is mainly distributed where the highest wind speeds of the region are observed, Fig 10l.

*C. hololeuca* was distributed over the whole region, Fig 8. As a consequence, there are no noticeable differences between the distributions of the climate variables extracted for *C. hololeuca* coordinates and extracted with random coordinates across the image, S1–S4 Figs.

## Discussion

### Mapping blooming trees

Here, a regional distribution map of all individuals of a natural tree species was produced based on its reproductive phenology using a U-net network and multispectral remote sensing images. The overall accuracy of 98.92% and an *IoU* of 0.867 could be explained by the unique spectral values (pink/purple) of the *T. pulchra* flowers, Fig 5. Other tropical tree species present intensely colored flowering blooms, such as the genus *Jacaranda* or *Tabebuia*, and could be mapped with U-net methods for ecological studies or to limit the damage of selective logging. Using flowering trees for mapping natural ecosystems has two main limitations. First, the number of adult trees is likely underestimated if the trees did not all flourish synchronously or every year such as already observed with the species *Tabebuia guayacan* at Barro Colorado Island [40]. Second, some other species can flower at the same time and have the same

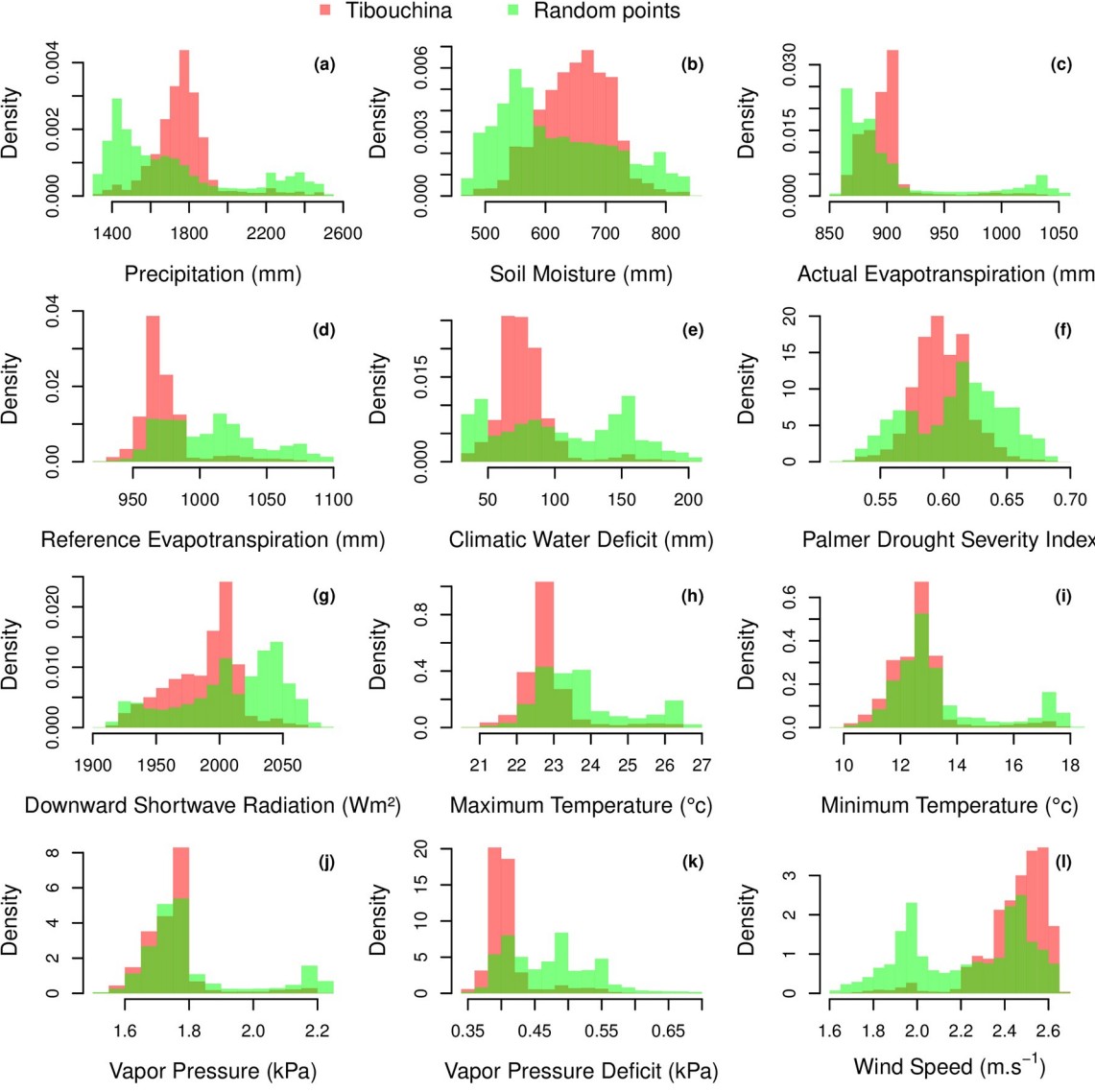

**Fig 10. Comparison of the distribution of the forests with or without *T. pulchra* dominance in relation with annual mean of climate variables over the Worldview2 image extent.** Climate variables were extracted for *T. pulchra* dominance polygon centroids (n = 461355) and from randomly distributed natural forests points without *T. pulchra* dominance (n = 50000).

reflectance in the satellite image. In the Atlantic forest, two other species of the genus *Tibouchina*, *T. mutabilis* and *T. sellowiana*, show a similar behaviour as *T. pulchra*. They flower the same time, show similar spectral values, and occur in the same secondary succession, independently of the species. They are all know under the same vernaculary name "Manacá da Serra". Due to this, we acknowledge that some errors in labelling can occur however all the mapped trees show similar characteristics that are to make pink/purple flowers synchronously and to dominate in new forests regenerating from pasture. Another problem with mapping flowering trees, specific with *T. pulchra*, is that they are also used as ornamental trees, and, consequently, they can be present outside of their natural or preferential range. There are some planted specimens at the National institute of Space Research of Brazil (INPE) and they are also sold on other continents. The method based on U-net and very high resolution images remains

difficult to apply on a biome scale for the flowering blooms as it would require an enormous amount of storage capacity and images without clouds at the blooming time. However, the signal of *T. pulchra* flowers might be found in the time series from high resolution image such as Landsat or Sentinel-2 data, and further work will be made to map the *T. pulchra* in all the Atlantic forest biome. *Tibouchina pulchra* flower at different moments of the year, earlier north of the Atlantic forest than in the south, and deciphering the date of flowering of the same species on gradients of precipitation, temperature and during several years and for different latitudes could help to understand the mechanisms and climate variables inducing flowering in tropical trees.

### Rio do Chapeu landscape history and its species indicators

The difference in forest cover between 1962 from the aerial image and 2017 from the U-net forest model over the Rio do Chapeu region showed that forest cover has largely increased, with more than half of the current forest cover (57%) grown after 1962, Fig 6. Among these new forests, almost one quarter were eucalyptus plantations, and only 20% of these plantations were made after the cut of an existing forest in 1962. After more than two centuries of intensive deforestation [6], this forest expansion is likely due to the federal law 5.106 of 1966 which has encouraged reforestation with fiscal incentive [19] by allowing farmers to apply 50% of their income tax to reforestation. It was also likely combined with the economical success of eucalyptus plantation in the early 1970s; in 1973, Brazil was already the largest eucalyptus producer in the world [41], and 80% of Brazil's plantations were located in the São Paulo state, where our region of interest is located [42].

Here, we find more than 90% of *T. pulchra* were inside or less than 50 m away from a new forest, Fig 8. They are only dominating in stands of new forests, after forest recovery from a pastureland, as already observed locally in a field study in this region [23]. It is not only its presence but rather its dominance that indicates forest regeneration on an abandoned pasture. Consequently, we can use this dominance information to assess the history of the landscape. While occurring mainly in new forests, *T. pulchra* does not occur in all the new forests. As we now know the recent history in this region, other indicators from remote sensing that can inform on the forest succession stage might be tested here, such as $\alpha$ and $\beta$ diversity computed directly from the image spectra [43, 44], canopy structure index based on textural information [45] or leaf area index, and carbon and nitrogen estimates from WorldView images [46]. *T. pulchra* population distribution is clearly related to regrowth after pasture; however, in a natural ecosystem, this dominant behaviour in open areas could be linked to landslides, which is the only natural process that can produce large opening in the forest that are comparable in size to pastureland, and which is the main process of landform development in this region [47].

While *Cecropia hololeuca* is one of the most common and recognisable tree in the Atlantic forest, our study shows a behaviour that was unknown before; that is, they occur mainly inside or close to old forest stands and rarely in new regeneration from pastureland. So they indicate canopy degradation of an old forest and not secondary succession from pasture. This was unexpected and not described before, likely because the individual needs to be mapped on a large scale to enable this observation, highlighting the importance of deep learning methods to improve our understanding of the tree species' biology. This distribution might have three main reasons: the first is that the *Cecropia hololeuca* germinated better in a forest gap or close to a forest border than in an open area. An alternative explanation could be that large, old individuals that are well detected by the u-net are only found inside or on the border of forest because this genus is sensitive to wind, with low wood density and low survival rate after

hurricanes such as observed in Puerto Rico by [48]. Finally, and more likely, this could be because of the previous land use. In the Amazon, it has been shown that *Cecropia sp.* occured in secondary forests after a clear cut only if there was no subsequent use of fire and conversion to pasture, and that another pioneer species, *Vismia sp.*, dominated after fire and pasture conversion [49, 50]. In the Atlantic forest, as our result shows, *C. hololeuca* trees occurred predominantly in old forests, and we might be in a similar scenario as in the Amazon; that is, the *C. hololeuca* are absent of the secondary forest succession after use of pasture and fire. In this Atlantic forest region, all pastures originated from the traditional slash and burn methods, made initially for coffee plantations that were further converted to pasture [6].

## Insights of large scale mapping for conservation and species distribution studies

The forest distribution of natural forest seems also to translate the human impacts. Remaining natural forest are present on steeper slopes and rougher terrain, which are obviously more difficult to exploit, and interestingly more oriented to the South, Fig 9. As reported by [6], in a manual for the coffee plantation management first published in 1847 by Francisco Peixoto de Lacerda Werneck and largely followed at the time, the plantation orientation on the South slopes was strongly discouraged [51]. This could explain why the natural forests with south orientation have been preserved until now. Based only on the species distribution, our results show that the forests of the region have undergone an important human influence, Fig 8. The forest degradation, described by the *C. hololeuca* dominance, represents 8.8% of the forest landscape. This dominance represents forests where the *C. hololeuca* concentration is above its mean concentration, and as our index of degradation is conservative, the degradation could be underestimated. Fortunately, our results also seem to indicate that these forests are old ($> 60$ years), and that they could have never been a pasture as *C. hololeuca* is present. This extends our previous findings [14] where, for the same WorldView-2 image, an index of disturbance based on *C. hololeuca* spatial distribution was produced, so now we know that these forests are old and which are the level of disturbance of these fragments. The regeneration of the forest, based on the *T. pulchra* dominance, represent 5% of the regional forest cover. Our results from the Rio do Chapeu, and because the 1966 law to encourage reforestation [19], seems to indicate that these new forests had grown after 1966. As not all new forests are dominated by *T. pulchra*, Fig 7, here we also underestimated the real regeneration from pasture. In Amazonia, abandoned clearcuts characterized by *Cecropia sp.* showed a richer mix of other arboreal genera characteristic of forest succession than arrested succession on abandoned pastures with history of fire dominated by the genus *Vismia*, another pioneer species [49]. Similarly, we observed C. hololeuca dominance mainly in old forests that have likely less or older history of fires than abandoned pasture where *T. pulchra* dominates. This could indicate that the forests characterized by the *C. hololeuca* dominance have a richer mix of species that the *Tibouchina* dominated forests but further studies combining biodiversity measured from the field, for example from the Biota Project [4], and our results are needed to confirm this. For the biodiversity studies and conservation, the dominance of *C. hololeuca* and *T. pulchra* measured from satellites could be important indices to report for the Atlantic forest.

The distribution of both species in relation to elevation and elevation-related variables show no remarkable differences with the distribution of the forest. This shows that analysis of forest distribution is necessary before studying individual species distribution to avoid biases. For example, in our case, the studies species are located on the south-oriented slopes, not because of their habitat preference, but because it is where the remaining/regrowing forests are. *C. hololeuca* is distributed over the region and did not show significant relation with

climate (Fig 1), as expected since its the large climate envelope covering all the Atlantic forest biome [52]. This is not the case for *T. pulchra* even at the regional scale. Its distribution relation with climate variable shows that this species seems to prefer wet climate with annual precipitation above $\sim$ 1600 mm and maximum annual temperature below 24˚ while its distribution seems unaffected by minimum temperature, Fig 10. However, we cannot conclude on the wettest part of the gradient because higher annual precipitation are observed in the south fragment, which has not undergone deforestation and agriculture, so even if there are natural occurrences of *T. pulchra* the natural conditions for dominance are limited. Further works are needed to develop methods to study tree species spatial distribution based on the map of all individuals.

## Conclusion

We have assessed the forest history in the 'Rio do Chapeu' region based on the difference of 1962 forest cover obtained from aerial images and of 2017 obtained through deep learning. The history of the forest was then related to the presence of two pioneer species: *Tibouchina pulchra*, which has been found to be dominant only in new forests regenerating from pasture, and *Cecropia hololeuca*, which is dominant only in old forests; that is, forests already existing in 1962 and with likely no preceding use of fire. The map of both species for the entire region shows that at least 5% of the current natural forests are recent regeneration from pasture, and that at least 10% of the natural forest show signs of important disturbance as indicated by the *C. hololeuca* dominance. Overall, we observed that the landscape has been highly impacted by humans. The positive point is that forest in this region has increased during the studied period, and that a lot of fragments are old forests (present before 1960 as deduced by the *C. hololeuca* dominance) likely containing more biodiversity that regeneration from pasture. The analysis of forest and species distribution show mainly: (i) that forests are oriented to the South and on steeper slopes that non forest, likely due to expansion of coffee plantation favoured on North slopes, (ii) that *Cecropia hololeuca* has a larger distribution range than the observed in studied region, and (iii) that *Tibouchina pulchra* seems to be restricted to wet environments with annual precipitation above 1600 mm. Finally, here we show how to use multiple sources of remote sensing data and new deep learning techniques to retrieve information on biology, such as tree species distribution and their behaviour, and how it can be used to interpret the landscape history.

## Supporting information

**S1 Fig. Distribution of the natural forests, excluding the largest fragment at the south, and non forest (drawn from a random distribution) in relation to elevation and five elevation-related variables at the regional scale.** The five elevation related indices consider the eight neighbor pixels and are slope (˚); aspect (compass direction facing slope in ˚, where 0 is the North); topographic position index (elevation of the pixel in relation to the mean elevation of the neighbour pixels); terrain ruggedness index (sum of the change in elevation between the pixels and its 8 neighbors cells); and, finally roughness (difference between the maximum and the minimum value of a cell and its 8 surrounding cells) [31, 32].
(PDF)

**S2 Fig. Comparison of the distribution of the forests with or without *T. pulchra* dominance in relation with elevation-related variables over the WorldView-2 image extent.** Elevation-related variables were extracted for *T. pulchra* dominance polygon centroids (n = 461355) and

from randomly distributed natural forests points without *T. pulchra* dominance (n = 50000).
(PDF)

**S3 Fig. Comparison of the distribution of the forests with or without *Cecropia hololeuca* dominance in relation with elevation and elevation-related variables over the WorldView-3 image extent.** Elevation-related variables were extracted for *C. hololeuca* dominance polygon centroids (n = 169542) and from randomly distributed natural forests points without *C. hololeuca* dominance (n = 11717).
(PDF)

**S4 Fig. Comparison of the distribution of the forests with or without *Cecropia hololeuca* dominance in relation with annual mean of climate variables over the WorldView-3 image extent.** Climate variables were extracted for *C. hololeuca* dominance polygon centroids (n = 169542) and from randomly distributed natural forests points without *C. hololeuca* dominance (n = 11717).
(PDF)

## Acknowledgments

We thank the two referees for their review and positive comments to improve the text. We thank DigitalGlobe for the provision of WorldView-2 and WorldView-3 satellite images.

## Author Contributions

**Conceptualization:** Fabien H. Wagner, Marcos P. M. Aidar, Yuliya Tarabalka.

**Data curation:** Fabien H. Wagner, André L. C. Rochelle.

**Formal analysis:** Fabien H. Wagner, Yuliya Tarabalka.

**Funding acquisition:** Fabien H. Wagner, Oliver L. Phillips, Emanuel Gloor, Luiz E. O. C. Aragão.

**Investigation:** Fabien H. Wagner, Marisa G. Fonseca.

**Methodology:** Fabien H. Wagner, Alber Sanchez, Yuliya Tarabalka.

**Project administration:** Fabien H. Wagner, Oliver L. Phillips, Emanuel Gloor.

**Resources:** Fabien H. Wagner.

**Software:** Fabien H. Wagner, Alber Sanchez.

**Supervision:** Fabien H. Wagner, Oliver L. Phillips, Emanuel Gloor, Luiz E. O. C. Aragão.

**Validation:** Fabien H. Wagner.

**Visualization:** Fabien H. Wagner.

**Writing – original draft:** Fabien H. Wagner.

**Writing – review & editing:** Alber Sanchez, Marcos P. M. Aidar, André L. C. Rochelle, Yuliya Tarabalka, Marisa G. Fonseca, Oliver L. Phillips, Emanuel Gloor, Luiz E. O. C. Aragão.

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
