## [Decision Letter · Decision Letter 0]

29 Nov 2019

PONE-D-19-23068

Mapping Atlantic rainforest degradation and regeneration history with indicator species using convolutional network

PLOS ONE

Dear Dr. Wagner,

Thank you for submitting your manuscript to PLOS ONE. After careful consideration, we feel that it has merit but does not fully meet PLOS ONE’s publication criteria as it currently stands. Therefore, we invite you to submit a revised version of the manuscript that addresses the points raised during the review process.

We would appreciate receiving your revised manuscript by Jan 13 2020 11:59PM. To enhance the reproducibility of your results, we recommend that if applicable you deposit your laboratory protocols in protocols.io, where a protocol can be assigned its own identifier (DOI) such that it can be cited independently in the future. For instructions see: http://journals.plos.org/plosone/s/submission-guidelines#loc-laboratory-protocols

We look forward to receiving your revised manuscript.

Kind regards,

Judi Hewitt

Academic Editor

PLOS ONE

Journal Requirements:

1. We note that you have stated that you will provide repository information for your data at acceptance. Should your manuscript be accepted for publication, we will hold it until you provide the relevant accession numbers or DOIs necessary to access your data. If you wish to make changes to your Data Availability statement, please describe these changes in your cover letter and we will update your Data Availability statement to reflect the information you provide.

Reviewers' comments:

Reviewer's Responses to Questions

**Comments to the Author**

1. Is the manuscript technically sound, and do the data support the conclusions?

Reviewer #1: Yes

Reviewer #2: Yes

2. Has the statistical analysis been performed appropriately and rigorously? 

Reviewer #1: Yes

Reviewer #2: Yes

3. Have the authors made all data underlying the findings in their manuscript fully available?

Reviewer #1: No

Reviewer #2: No

4. Is the manuscript presented in an intelligible fashion and written in standard English?

Reviewer #1: Yes

Reviewer #2: Yes

5. Review Comments to the Author

Reviewer #1: This is a highly technical paper and I am not an expert in remote sensing techniques, so I cannot really judge all of the methods. That said, the approach seems very rigorous to me and offers new insights into species distributions and how they can be detected using high-resolution remote sensing data. The approach is focused on classifying two tree species that are "indicator" species of young and old secondary forest in the region. The results on regrowth forest and Eucalyptus plantations is very interested and relevant. The limitations of the methods are clearly presented, but it is not clear how errors due to including other flowering species might change the results of the analysis.

page 12, line 11: As there are only two time periods assessed, how do you know that forests directly converted to plantations? There could have been intermediate agricultural land use.

Section 5.1: how would results be affected if some of these other flowering trees were labelled as Tibouchina pulchra? Are they also good indicators of regenerating forests?

page 21, line 3: Most pastures in São Paulo state were derived from coffee plantations and not from shifting cultivation.

page 21, line 24: Where are these data presented? I don't think that evidence is presented in this paper to support this claim. Results from Amazonia in Cecropia vs. Vismia dominated secondary forests are not relevant here.

page 22, line 11: these still could be old second-growth (more than 60 yr old). Deforestation happened centuries ago in this region. Old forest is a vague term and for some it means old-growth forest (not cleared historically).

Reviewer #2: General Comments

The authors present in this work a method to evaluate landscape forest degradation and regeneration history using very high resolution satellite data and aerial photographs to map the occurrence of two indicator species. The method is based on machine learning and identifies the presence of species based on the image’s spectral characteristics. They have also crossed this species occurrence data to local climate and geomorphological data in order to understand the species’ habitat preferences.

The workflow of the geoprocessing is well described and includes a good error estimation section, which increases the robustness of the results. This is important because in my view the framework has good potential for expansion to larger areas and use as a tool for other studies, especially in areas in which Cecropia hololeuca and Tibouchina pulchra occur in the netropics, but also by using other local species. However, in order to increase the usage of this method by the scientific community, some aspects especially regarding scientific reproducibility should be included.

First, scientific research with remote sensing is heavily based on freely available satellite data, which is not the case with the very high resolution Worldview catalogue. It would be invaluable to analyze how free datasets (e.g. Landsat, CBERS, Sentinel-2) fare in comparison to the Worldview data for detecting the target species within the framework (even if they perform poorly, strengthening the case for free very high resolution data). Second, it is important that the algorithm code becomes available through GitHub or another code sharing platform from the time of the publication, together with an image sample from 1962 and another from Worldview, providing a demo of the analysis. This would strengthen the citation and usage frequency by this study.

Finally, I noticed in this study a general lack of usage of forest fragmentation as an analysis and discussion base within tropical ecology (Laurance et al. 2011; Chaplin-Kramer et al. 2015; Lôbo et al. 2011) and remote sensing (Dantas De Paula, Groeneveld, and Huth 2016; Pettorelli 2015; Melito, Metzger, and de Oliveira 2018). This I feel would be invaluable for understanding the occurrence of the indicator species within a forest edge-core gradient.

Specific comments:

Page 1, Line 7 – elevation data instead of only elevation.

P2L4 – Also high diversity forest restoration initiatives in the Atlantic Forest, see (Melo et al. 2013; Rodrigues et al. 2011)

P2L24- would eucalyptus plantations be considered as reforestation?

P2L29 – Maybe not relevant to include common names.

P3L29 - I know that in Brazil the forest domains are called "biomes" but this has another connotation in the outside scientific community, maybe use another term?

P5L4 – The digitalized images are panchromatic or have also RGB bands?

P5L27 - Since SRTM is a surface model and not a terrain model, ruggedness could be influenced by canopy characteristics.

P8L2 - How does this compare with conventional supervised/unsupervised image classification methods?

P10L5 – I suggest that a sample analysis demo with the complete code be prepared and shared for this study.

P10L9 - Why not also use other low resolution forest change maps in addition (e.g. Hansen et al. 2013)

P12L8 - The data should be better presented, in order to show how forest cover (natural and plantations) has changed (i.e. loss or gained) from 1962 to 2017.

P12L14 - Probably the fact that these remaining forests are now edge-dominated fragments?

Fig.6 - This could be better presented with a "forest change" color scheme similar to Hansen (unchanged, gain, loss in relation to 1962).

Fig.7 – Same as Fig. 6.

P17L7 - Would be interesting to see how the species are located in terms of edge distance.

P20L25 - This may be related to fragmental collapse.

References

Chaplin-Kramer, Rebecca, Ivan Ramler, Richard Sharp, Nick M Haddad, James S Gerber, Paul C West, Lisa Mandle, et al. 2015. “Degradation in Carbon Stocks near Tropical Forest Edges.” Nature Communications 6: 10158. https://doi.org/10.1038/ncomms10158.

Dantas De Paula, Mateus, Jürgen Groeneveld, and Andreas Huth. 2016. “The Extent of Edge Effects in Fragmented Landscapes: Insights from Satellite Measurements of Tree Cover.” Ecological Indicators 69: 196–204. https://doi.org/10.1016/j.ecolind.2016.04.018.

Hansen, M. C., P. V. Potapov, R. Moore, M. Hancher, S. A. Turubanova, A. Tyukavina, D. Thau, et al. 2013. “High-Resolution Global Maps of 21st-Century Forest Cover Change.” Science 342 (6160): 850–53. https://doi.org/10.1126/science.1244693.

Laurance, William F., José L.C. Camargo, Regina C.C. Luizão, Susan G Laurance, Stuart L Pimm, Emilio M Bruna, Philip C Stouffer, et al. 2011. “The Fate of Amazonian Forest Fragments: A 32-Year Investigation.” Biological Conservation. https://doi.org/10.1016/j.biocon.2010.09.021.

Lôbo, Diele, Tarciso Leão, Felipe P L Melo, André M M Santos, and Marcelo Tabarelli. 2011. “Forest Fragmentation Drives Atlantic Forest of Northeastern Brazil to Biotic Homogenization.” Diversity and Distributions 17 (2): 287–96. https://doi.org/10.1111/j.1472-4642.2010.00739.x.

Melito, Melina, Jean Paul Metzger, and Alexandre A. de Oliveira. 2018. “Landscape-Level Effects on Aboveground Biomass of Tropical Forests: A Conceptual Framework.” Global Change Biology 24 (2): 597–607. https://doi.org/10.1111/gcb.13970.

Melo, Felipe P.L., Víctor Arroyo-Rodríguez, Lenore Fahrig, Miguel Martínez-Ramos, and Marcelo Tabarelli. 2013. “On the Hope for Biodiversity-Friendly Tropical Landscapes.” Trends in Ecology & Evolution 28 (8): 462–68. https://doi.org/10.1016/j.tree.2013.01.001.

Pettorelli, Nathalie. 2015. “Agree on Biodiversity Metrics to Track from Space.” Nature 523: 5–7. https://doi.org/10.1038/523403a.

Rodrigues, Ricardo Ribeiro, Sergius Gandolfi, André Gustavo Nave, James Aronson, Tiago Egydio Barreto, Cristina Yuri Vidal, and Pedro H.S. Brancalion. 2011. “Large-Scale Ecological Restoration of High-Diversity Tropical Forests in SE Brazil.” Forest Ecology and Management 261 (10): 1605–13. https://doi.org/10.1016/j.foreco.2010.07.005.

6. PLOS authors have the option to publish the peer review history of their article (what does this mean?). If published, this will include your full peer review and any attached files.

Reviewer #1: No

Reviewer #2: Yes: Mateus Dantas de Paula

---

## [Author Response · Author response to Decision Letter 0]

20 Jan 2020

Response to Reviewers

5. Review Comments to the Author

Reviewer #1: This is a highly technical paper and I am not an expert in remote sensing techniques, so I cannot really judge all of the methods. That said, the approach seems very rigorous to me and offers new insights into species distributions and how they can be detected using high-resolution remote sensing data. The approach is focused on classifying two tree species that are "indicator" species of young and old secondary forest in the region. The results on regrowth forest and Eucalyptus plantations is very interested and relevant. The limitations of the methods are clearly presented, but it is not clear how errors due to including other flowering species might change the results of the analysis.

FW : Dear Reviewer #1, thank you very much for you positive review that I believe have improved the paper. In the following text you will find your comments and our answers in blue.

page 12, line 11: As there are only two time periods assessed, how do you know that forests directly converted to plantations? There could have been intermediate agricultural land use.

FW : You are right it could have been intermediate use. To clarify the sentence was changed to: “The eucalyptus were mainly planted where there was no forest in 1962; only 21.1 % of the plantation were natural forests in 1962. As only two time periods have been assessed, 1962 and 2017, intermediate land uses that could have existed between transition from natural forests and to planted forests are unknown.”

Section 5.1: how would results be affected if some of these other flowering trees were labelled as Tibouchina pulchra? Are they also good indicators of regenerating forests?

FW: You are right some other flowering trees can be labelled Tibouchina pulchra, likely other Tibouchina species. However, from the results, we find that ”66.8% of these trees were located in new forests that were pasture in 1962, 81.4% inside or at a distance below 25 m of a new forest and 89.7% inside or at a distance below 50 m of a new forest”. So even considering errors in the labelling, we find a similar preference for the habitat, that is inside or nearby a regenerating forest, for all the mapped trees. Furthermore, there is a clear dominance behaviour of the mapped species: “in the neighbouring area of 25 m around each Tibouchina pixels, there is a mean of 17.5 % of Tibouchina pixels (343.5 m²)”; so even considering labelling errors this dominance behaviour seems also shared. And, we mapped regeneration only based on the dominance of the trees, which means that our method does not account for isolated individuals. Finally, the massive and time synchronous flowering with a similar spectral value is shared among all this trees. To conclude, even some errors in labels can occurs, all the mapped trees show similar characteristics that are (i) to make flowers synchronously and (ii) to dominate in new forests regenerating from pasture. 

In the text, the following sentence was added: “ … are all know under the same vernaculary name "Manacá da Serra". Due to this, we acknowledge that some errors in labelling can occur however all the mapped trees show similar characteristics that are to make pink/purple flowers synchronously and to dominate in new forests regenerating from pasture.”

page 21, line 3: Most pastures in São Paulo state were derived from coffee plantations and not from shifting cultivation.

FW: I totally agreed with that, the main transition was likely: forest -> coffee plantations -> pastures. I think it is exactly the meaning of the sentence “In this Atlantic forest region, all pastures originated from the traditional slash and burn methods, made initially for coffee plantations that were further converted to pasture (Dean, 1997).”. However, here Dean does no refer to slash and burn cultivation but only to slash and burn technics to prepare the land for coffee plantation, but I understand it can be a bit confusing for the reader. To clarify the sentence was changed to “In this Atlantic forest region, all pastures originated from forest cutting and burning, made initially for coffee plantations that were further converted to pasture (Dean, 1997).”.

page 21, line 24: Where are these data presented? I don't think that evidence is presented in this paper to support this claim. Results from Amazonia in Cecropia vs. Vismia dominated secondary forests are not relevant here.

FW : The paper of Cecropia vs. Vismia show that the forest succession is different depending on the past use of fire, that is, if only clearcut, Cecropia sp. dominated, while in case of clearcut + fire, Vismia sp. dominated. I really believe that it could be linked at some point to what we are observing in the Atlantic forest, that is C. hololeuca are almost absent in the succession from pasture regeneration. This is the closest study have found to discuss my results. I have changed the sentence to “Similarly, we observed C. hololeuca dominance mainly in old forests that have likely less or older history of fires than abandoned pasture where T. pulchra dominates. This could indicate that the forests characterized by the C. hololeuca dominance have a richer mix of species that the Tibouchina dominated forests but further studies combining biodiversity measured from the field, for example from the BiotaProject (Joly et al., 2014), and our results are needed to confirm this.” 

page 22, line 11: these still could be old second-growth (more than 60 yr old). Deforestation happened centuries ago in this region. Old forest is a vague term and for some it means old-growth forest (not cleared historically).

FW: Yes, you are right, and, in the sentence, I have removed “and with likely no preceding use of ﬁre.” which was more an assumption that have emerged from the discussion than a result. The term old forest in the paper refer only to forest present in 1962 as defined in the section 3.2.1 of the Methods. And to clarify, I sometimes repeat the paper definition of old forest. I didn’t find a better term, and for the graphics it is better as it is short and very discriminative, but if you have a suggestion, I can include it.

Just as a remark, in my opinion these forests are very old and have at least more than a century. In the aerial photographs, I have saw that there was already Cecropias in almost all the fragments with Cecropias today, that how I had the idea of this study, that is that this species only occurs inside old forest fragments and that’s what our results shows (and was different of what botanist were thinking of this species). I didn’t analyse this because it will be too qualitative: the quality of the aerial photography is not so good as satellite image and I only have images from the Rio do Chapeu region, however, you can clearly see some white spots in the fragments which are the Cecropias, see Fig. 1. So, following our results, this means that these fragments were forest at least 60 years before the aerial photographs of 1962. 

Figure 1 : Example of forest fragments with visible Cecropias in the 1962 aerial image from the rio do chapeu region (these fragments can be found in the figure 7 of the paper in the top left part)

Reviewer #2: General Comments

The authors present in this work a method to evaluate landscape forest degradation and regeneration history using very high resolution satellite data and aerial photographs to map the occurrence of two indicator species. The method is based on machine learning and identifies the presence of species based on the image’s spectral characteristics. They have also crossed this species occurrence data to local climate and geomorphological data in order to understand the species’ habitat preferences.

The workflow of the geoprocessing is well described and includes a good error estimation section, which increases the robustness of the results. This is important because in my view the framework has good potential for expansion to larger areas and use as a tool for other studies, especially in areas in which Cecropia hololeuca and Tibouchina pulchra occur in the netropics, but also by using other local species. However, in order to increase the usage of this method by the scientific community, some aspects especially regarding scientific reproducibility should be included.

First, scientific research with remote sensing is heavily based on freely available satellite data, which is not the case with the very high resolution Worldview catalogue. It would be invaluable to analyze how free datasets (e.g. Landsat, CBERS, Sentinel-2) fare in comparison to the Worldview data for detecting the target species within the framework (even if they perform poorly, strengthening the case for free very high resolution data). Second, it is important that the algorithm code becomes available through GitHub or another code sharing platform from the time of the publication, together with an image sample from 1962 and another from Worldview, providing a demo of the analysis. This would strengthen the citation and usage frequency by this study.

Finally, I noticed in this study a general lack of usage of forest fragmentation as an analysis and discussion base within tropical ecology (Laurance et al. 2011; Chaplin-Kramer et al. 2015; Lôbo et al. 2011) and remote sensing (Dantas De Paula, Groeneveld, and Huth 2016; Pettorelli 2015; Melito, Metzger, and de Oliveira 2018). This I feel would be invaluable for understanding the occurrence of the indicator species within a forest edge-core gradient.

FW : Dear Reviewer #2, thank you very much for you positive review, I have try to account of most of you comment and believe it has improved the paper.

I have now added the data of Cecropia and Tibouchina dominances and natural forest cover (Figure 8) in shapefile as well as the U-net model code example with a simulated WordView image on Zenodo public repository(map data: https://doi.org/10.5281/zenodo.3601487 and the Unet model example : https://doi.org/10.5281/zenodo.3601503 ). Unfortunately, I can’t share the original Worldview and the aerial images as I don’t have the rights to do this.

I understand your point of using other dataset, but it goes beyond the scope of this paper, that is why this appears in perspectives in the discussion. I am currently working on this with sentinel data, it works at some point, but you definitively need high resolution image to be 100% sure to identify the objects you are mapping. And working with low resolution goes with tons of other problem such as registration, shade, clouds, no data etc… I have schedule one entire year to work on this, it is not something that can be done quickly and added in this paper. This will be the next work and I am convinced there are some very promising perspectives for large scale species mapping.

The fragmentation analysis you are asking to me, based on the same Cecropia map used in this work, has been published in Remote Sensing in Ecology and Conservation with Nathalie Pettorelli as Editor in 2019 (see here: https://zslpublications.onlinelibrary.wiley.com/doi/full/10.1002/rse2.111). It includes a more detailed methodology section as well as the complete analysis of the Cecropia distance to the edge in all the fragments, plus a discussion with full section focused on fragmentation. That why I didn’t much explore this in this paper and focus more on the history of the fragment. For the Tibouchina this analysis is not so relevant because they are dominant, so they are distributed all over the new forests, from the edge to the center. I didn’t make this distribution analysis because I know that it will bring more work than valuable information. In my opinion, the best information that Tibouchina trees can show us is that a fragment dominated by Tibouchina is a recent regeneration of a pasture, which is, I believe a new and very valuable information for conservation. 

Furthermore, in the other paper I have computed an index for each fragment but what I have saw that there is a problem to deal with large fragment and you lost important information, that is why I have adopted a more pixel based approach. For example, considering a large fragment of 1000 ha with 8 hectares of disturbed forest with Cecropia dominance, if I compute one disturbance index, let’s say the percentage of Cecropia dominance, the fragment will be classified as undisturbed because Cecropia dominance only represent 0.8 %. Now if we consider a fragment of 10 with the same 8 hectares of disturbed forest with Cecropia dominance, the fragment will be now classified as disturbed because Cecropia dominance represent 80% of the fragment. In both case we have 8 ha dominated by Cecropia and in the case of large fragment we totally lost this information. With the method developed in this paper, independently of the fragment size, we can map exactly where the disturbance is and its size, it goes beyond fragmentation analysis, we are looking at indicator species distribution inside the fragment to identify forest degradation history, something that nobody have done before at this scale with high resolution image. 

In the following text you will find your comments and our answers in blue.

Specific comments:

Page 1, Line 7 – elevation data instead of only elevation.

FW : agree and corrected

P2L4 – Also high diversity forest restoration initiatives in the Atlantic Forest, see (Melo et al. 2013; Rodrigues et al. 2011)

FW : agree and added. I have only added the reference of Rodrigues et al because in contains real data while the Melo et al paper is more about concepts and potential implications of successional trajectories for conservation planning. The sentence now reads :” This increase in tree cover is driven mainly by eucalyptus plantations and natural regeneration (Silva et al., 2018), and also by forest restoration initiatives (Rodrigues et al., 2011).”

P2L24- would eucalyptus plantations be considered as reforestation ?

FW : you are right, plantation are not considered as reforestation but as afforestation, the sentence now reads “… after 1966, when the Brazilian federal law 5.106 came into force with the goal to encourage afforestation and reforestation in Brazil (Sampaio, 1975).”

P2L29 – Maybe not relevant to include common names.

FW : I still think that this is a nice information, particularly for the Brazilian readers who know only this tree by the common name. For example, when I present this work to Brazilians and people working on Atlantic forest, a large part of the audience know this species with the vernaculary name “Manaca da serra” as this species is really common in the Atlantic forest, but if I say only Tibouchina pulchra, nobody knows which plant it is.

P3L29 - I know that in Brazil the forest domains are called "biomes" but this has another connotation in the outside scientific community, maybe use another term?

FW : This is the term in use in our scientific community, and also used as a synonym of ‘domain’ (which I believe is term you are suggesting). For example, the term ‘biome’ is used in all the publications about Atlantic forest you cite in references and used as a synonym of ‘domain’ in previous Plos One article on Atlantic forest (for example Bogoni et al, 2018, https://doi.org/10.1371/journal.pone.0204515). So, as there is no clear consensus on using biome or domains I have kept biome in the text for consistency. 

Bogoni JA, Pires JSR, Graipel ME, Peroni N, Peres CA (2018) Wish you were here: How defaunated is the Atlantic Forest biome of its medium- to large-bodied mammal fauna? PLoS ONE 13(9): e0204515. https://doi.org/10.1371/journal.pone.0204515

P5L4 – The digitalized images are panchromatic or have also RGB bands?

FW : The aerial images are panchromatic, so they look as a gray image (see Figure 1 above, in the responses to Reviewer#1). The digital images of the aerial images are *.png, so they have RGB bands, however the image still looks as a gray image. In the text I have now included this information: “In 1962, ten years before the launch of the first Landsat satellite, panchromatic aerial photographs were taken over São Paulo…”

P5L27 - Since SRTM is a surface model and not a terrain model, ruggedness could be influenced by canopy characteristics.

FW : As the spatial resolution of SRTM is of 30 m and the ruggedness computed on a moving windows of 3x3 pixel (90m), I don’t expect so much influence of the canopy characteristics and the forest region is really not flat, more like a succession of hills, so on 90 m, I expect more variation of elevation from the terrain that from the canopy.

P8L2 - How does this compare with conventional supervised/unsupervised image classification methods?

FW : It is important to point out that the result of the U-net model already gives the segmentation of the trees. In traditional non-deep learning approaches, we can separate two methods that could be used for the species identification, object-based image analysis or pixel-based analysis. Both methods need more than one step to produce the species map.

First, pixels-based analysis will likely fail because in the crown of Cecropia or Tibouchina there is a large variability of reflectance, see Figure 5 of the paper for Tibouchina (or Figure 3 of of the paper https://zslpublications.onlinelibrary.wiley.com/doi/full/10.1002/rse2.111 for the Cecropia). For example, in the Cecropia crowns there are bright pixels and other are darks. This is due to the architecture of the tree. Even if the method could find the bright pixels that we could attribute to the Cecropia, then we would have to find a method to group them as a crown, which mean a human based threshold, which is likely more subjective that the U-net approach, and could never reach the accuracy of Unet.

Second, for object-based image analysis, we would also have to make two steps, the first is the segmentation of the individual tree crowns (ITCs) and the second is the identification of the species of these crowns. There is no non-deep learning individual tree crown segmentation method that can reach level of accuracy above 90%, even with hyperspectral and lidar data combined [Tochon et al., 2015; Dalponte et al., 2014; Singh et al., 2015]. Furthermore, Dalponte et al. (2014) reported that large and small crowns density tend to be underestimated when optical images are used to delineate ITCs. And after this uncertain tree crown segmentation, it remains to predict the species. There are absolutely no chances for a non-deep learning algorithm to reach accuracies obtained here with the U-net model for image segmentation, and this has already been demonstrated by the work of [Huang et al., 2018], paper which is cited in introduction.

References :

M. Dalponte, H. O. Orka, L. T. Ene, T. Gobakken, E. Naesset, Tree crown delineation and tree species classi_cation in boreal forests using hyperspectral and ALS data, Remote Sensing of Environment 140 (2014) 306-317.

G. Tochon, J. Fret, S. Valero, R. Martin, D. Knapp, P. Salembier, J. Chanussot, G. Asner, On the use of binary partition trees for the tree crown segmentation of tropical rainforest hyperspectral images, Remote Sensing of Environment 159 (2015) 318-331.

M. Singh, D. Evans, B. S. Tan, C. S. Nin, Mapping and characterizing selected canopy tree species at the angkor world heritage site in Cambodia using aerial data, PLOS ONE 10 (2015) 1-26.

Huang, B.; Lu, K.; Audebert, N.; Khalel, A.; Tarabalka, Y.; Malof, J.; Boulch, A.; Le Saux, B.; Collins, L.; Bradbury, K.; Lefèvre, S.; El-Saban, M. Large-scale semantic classification: outcome of the first year of Inria aerial image labeling benchmark. IEEE International Geoscience and Remote Sensing Symposium – IGARSS 2018.

P10L5 – I suggest that a sample analysis demo with the complete code be prepared and shared for this study.

FW : agreed and done. The U-net code for training and prediction with a simulated WorldView image have been made available on a public repository here: https://doi.org/10.5281/zenodo.3601503. I also have made available the maps of Cecropia and Tibouchina dominances and natural forest cover here : https://doi.org/10.5281/zenodo.3601487

P10L9 - Why not also use other low resolution forest change maps in addition (e.g. Hansen et al. 2013)

FW : Mainly because (i) the WorldView image is not orthorectified so depending on the elevation and the satellite view angle the low and high resolution does not match spatially very well, I will have to orthorectified the WorldView images first to match the low resolution data which is a lot of work and in comparison of making my own map and (ii) I can compute the accuracy on my own map, while we don’t know so much about the Hansen data accuracy for this region and finally (iii) low resolution with 30 m will lead to more error on estimation of the forest fragment size, with high resolution I can do far better.

P12L8 - The data should be better presented, in order to show how forest cover (natural and plantations) has changed (i.e. loss or gained) from 1962 to 2017.

FW : It is not tree cover change (such as previously written in the title of section 3.2.1 but have been corrected now) but only the past history of the forest mapped in 2017. It is the difference between the polygons of forests in 2017 and the polygons of forests in 1962 that were still forests in 2017. To map the past forests, I have made a first mask of the forest in 2017 and then only delineate forests in 1962 inside this mask. So there is only 3 classes, which are the classes of the Figure 6: old forests, new forests, and eucalyptus forest. And, all Eucalyptus plantation of the region were all made after 1962. That was already a huge work of manually drawing the past forest segments and I didn’t do that for loss because it was out of the scope of this paper, which is more about determining the degradation history of exiting forest fragments. I have now rewritten the method and change the name of the section 3.2.1 to clarify this. The paragraph is now:

“3.2.1 Past history of the 2017 forest cover

First, the mask of natural/planted forest in 2017 was produced with the U-net at 0.5 m. Second, the past history of the 2017 forest cover, that is, if they were present or not in 1962, was mapped. To do this, only for the forests that were mapped in 2017, the mask of the forests in 1962 was produced with the registered aerial images of 1962. This map was produced in QGIS and resampled in a raster tile at 0.5 m spatial resolution (QGIS Development Team, 2009). The history of the 2017 forest cover was then described in the "Rio do Chapeu" region. With these two maps, we were ……”.

P12L14 - Probably the fact that these remaining forests are now edge-dominated fragments?

FW : sorry but I am not sure to understand this comment. The result P12L14 is to compare to the Tibouchina results: ~ 90% of the Cecropia are located inside or a at maximum of 50 m of an old forest and ~90% of the Tibouchina are located inside or at a maximum of 50 m of a new forest.

Fig.6 - This could be better presented with a "forest change" color scheme similar to Hansen (unchanged, gain, loss in relation to 1962).

FW : As in the response of one of your previous comment, the classes are not forest change but only past history of the forests existing in 2017. I have corrected the paragraph 3.2.1 to clarify this.

Fig.7 – Same as Fig. 6.

FW : Same as previous comment.

P17L7 - Would be interesting to see how the species are located in terms of edge distance.

FW : The complete analysis of the distance to the edge with the same Cecropia map has been published in 2019 in Remote Sensing in Ecology in Conservation, see here: https://zslpublications.onlinelibrary.wiley.com/doi/full/10.1002/rse2.111

P20L25 - This may be related to fragmental collapse.

FW: Yes, it might but unfortunately, I don’t have the results to support this assumption. Could be great to have LiDAR data of the fragment to test for this.

References

Chaplin-Kramer, Rebecca, Ivan Ramler, Richard Sharp, Nick M Haddad, James S Gerber, Paul C West, Lisa Mandle, et al. 2015. “Degradation in Carbon Stocks near Tropical Forest Edges.” Nature Communications 6: 10158. https://doi.org/10.1038/ncomms10158.

Dantas De Paula, Mateus, Jürgen Groeneveld, and Andreas Huth. 2016. “The Extent of Edge Effects in Fragmented Landscapes: Insights from Satellite Measurements of Tree Cover.” Ecological Indicators 69: 196–204. https://doi.org/10.1016/j.ecolind.2016.04.018.

Hansen, M. C., P. V. Potapov, R. Moore, M. Hancher, S. A. Turubanova, A. Tyukavina, D. Thau, et al. 2013. “High-Resolution Global Maps of 21st-Century Forest Cover Change.” Science 342 (6160): 850–53. https://doi.org/10.1126/science.1244693.

Laurance, William F., José L.C. Camargo, Regina C.C. Luizão, Susan G Laurance, Stuart L Pimm, Emilio M Bruna, Philip C Stouffer, et al. 2011. “The Fate of Amazonian Forest Fragments: A 32-Year Investigation.” Biological Conservation. https://doi.org/10.1016/j.biocon.2010.09.021.

Lôbo, Diele, Tarciso Leão, Felipe P L Melo, André M M Santos, and Marcelo Tabarelli. 2011. “Forest Fragmentation Drives Atlantic Forest of Northeastern Brazil to Biotic Homogenization.” Diversity and Distributions 17 (2): 287–96. https://doi.org/10.1111/j.1472-4642.2010.00739.x.

Melito, Melina, Jean Paul Metzger, and Alexandre A. de Oliveira. 2018. “Landscape-Level Effects on Aboveground Biomass of Tropical Forests: A Conceptual Framework.” Global Change Biology 24 (2): 597–607. https://doi.org/10.1111/gcb.13970.

Melo, Felipe P.L., Víctor Arroyo-Rodríguez, Lenore Fahrig, Miguel Martínez-Ramos, and Marcelo Tabarelli. 2013. “On the Hope for Biodiversity-Friendly Tropical Landscapes.” Trends in Ecology & Evolution 28 (8): 462–68. https://doi.org/10.1016/j.tree.2013.01.001.

Pettorelli, Nathalie. 2015. “Agree on Biodiversity Metrics to Track from Space.” Nature 523: 5–7. https://doi.org/10.1038/523403a.

Rodrigues, Ricardo Ribeiro, Sergius Gandolfi, André Gustavo Nave, James Aronson, Tiago Egydio Barreto, Cristina Yuri Vidal, and Pedro H.S. Brancalion. 2011. “Large-Scale Ecological Restoration of High-Diversity Tropical Forests in SE Brazil.” Forest Ecology and Management 261 (10): 1605–13. https://doi.org/10.1016/j.foreco.2010.07.005.

---

## [Editor Report · Decision Letter 1]

7 Feb 2020

Mapping Atlantic rainforest degradation and regeneration history with indicator species using convolutional network

PONE-D-19-23068R1

Dear Dr. Wagner,

We are pleased to inform you that your manuscript has been judged scientifically suitable for publication and will be formally accepted for publication once it complies with all outstanding technical requirements.

With kind regards,

Judi Hewitt

Academic Editor

PLOS ONE
---

## [Editor Report · Acceptance letter]

10 Feb 2020

PONE-D-19-23068R1 

Mapping Atlantic rainforest degradation and regeneration history with indicator species using convolutional network 

Dear Dr. Wagner:

I am pleased to inform you that your manuscript has been deemed suitable for publication in PLOS ONE. Congratulations! Your manuscript is now with our production department. 

With kind regards,

on behalf of

Dr. Judi Hewitt 

Academic Editor

PLOS ONE